# Temporal dendritic heterogeneity incorporated with spiking neural networks for learning multi-timescale dynamics

Hanle Zheng [1], Zhong Zheng[1], Rui Hu[1], Bo Xiao[1], Yujie Wu[2], Fangwen Yu[1], Xue Liu[1], Guoqi Li[3] & Lei Deng [1] ✉

It is widely believed the brain-inspired spiking neural networks have the capability of processing temporal information owing to their dynamic attributes. However, how to understand what kind of mechanisms contributing to the learning ability and exploit the rich dynamic properties of spiking neural networks to satisfactorily solve complex temporal computing tasks in practice still remains to be explored. In this article, we identify the importance of capturing the multi-timescale components, based on which a multi-compartment spiking neural model with temporal dendritic heterogeneity, is proposed. The model enables multi-timescale dynamics by automatically learning heterogeneous timing factors on different dendritic branches. Two breakthroughs are made through extensive experiments: the working mechanism of the proposed model is revealed via an elaborated temporal spiking XOR problem to analyze the temporal feature integration at different levels; comprehensive performance benefits of the model over ordinary spiking neural networks are achieved on several temporal computing benchmarks for speech recognition, visual recognition, electroencephalogram signal recognition, and robot place recognition, which shows the best-reported accuracy and model compactness, promising robustness and generalization, and high execution efficiency on neuromorphic hardware. This work moves neuromorphic computing a significant step toward real-world applications by appropriately exploiting biological observations.

Inspired by the structures and functions of neural circuits in the brain, spiking neural networks (SNNs) are modeled and known as the third-generation artificial neural networks (ANNs)[1]. The studies of SNNs have considered abundant biological observations in terms of neural dynamics, connection patterns, coding schemes, processing flows, and so forth. In recent years, SNNs have gained extensive attention in image recognition tasks[2–4], especially after the boost of accuracy by borrowing the backpropagation through time (BPTT) learning algorithm from the ANN domain[5]. Owing to the dynamic attributes of

SNNs, they are widely believed to have the capability of processing temporal information. However, how to understand what kind of mechanisms contributing to the learning ability and exploit the rich dynamic properties of SNNs to satisfactorily solve complex temporal computing tasks still remain to be explored.

We rethink the key capabilities required in performing real-world temporal computing tasks. Complex temporal signals usually present variable timescales and high spectral richness, which can be well handled by the brain[6]. For example, the brain can easily recognize

[1]Center for Brain Inspired Computing Research (CBICR), Department of Precision Instrument, Tsinghua University, Beijing, China. [2]Institute of Theoretical Computer Science, Graz University of Technology, Graz, Austria. [3]Institute of Automation, Chinese Academy of Sciences, Beijing, China. ✉e-mail: leideng@mail.tsinghua.edu.cn

speakers who are speaking with different timescales such as fast or slow. Unlike the mainstream image recognition benchmarks used by SNNs based on static images and dynamic events collected by dynamic vision sensors (DVS)[7,8], the information in temporal computing tasks often shows complicated temporal dependencies, and the features appear with various timescales, which imply that the learning of multi-timescale temporal dynamics might be a critical point. In essence, neuroscientists have observed huge temporal heterogeneity in brain circuits and responses[9,10], for example, neural heterogeneity[11,12], dendritic heterogeneity[13–15] and synaptic heterogeneity[16–18]. It seems believable that these kinds of heterogeneity are more than noises but promising to generate the capability of capturing and processing multi-timescale temporal features[19]. While the investigation of synaptic heterogeneity offers valuable insights, it poses significant challenges in network modeling due to the high computation and storage overhead with the vast number of synapses. Furthermore, we found that only considering neural heterogeneity makes it hard to deliver satisfactory results when performing temporal computing tasks due to the insufficient multi-timescale neural dynamics. In light of these limitations, our work focuses on the exploration of dendrite heterogeneity as a more effective and efficient alternative in practice.

Computational neuroscientists have paid attention to the temporal computing capabilities of dendrites inferred from many biophysical phenomena and proposed neuron models[20–24] or fabricated dendrite-like nanoscale devices[25,26] to mimic biological behaviors. The advanced computational functions suggested by biological dendrites including local nonlinear transformation[20,27], adjustment to synaptic learning rules[22,28], multiplexing different sources of neural signals[29] and the generation of multi-timescale dynamics[13,23] may benefit neural networks in machine learning. Whereas, these biological observations are hard to apply to real-world temporal computing tasks performed with neural networks at the current stage due to the inappropriate abstraction, the high computational complexity and the lack of effective learning algorithms. In addition, most of existing SNNs for solving real-world temporal computing tasks adopt the simplified version of leaky integrate-and-fire (LIF) neurons[30], which cannot sufficiently exploit the rich temporal heterogeneity. Even though a few researchers such as Perez-Nieves et al.[31] have touched the neural heterogeneity by learning membrane and synaptic time constants, they ignored the dendritic heterogeneity which we consider of great importance. Recently, some researchers have noticed it and tried to develop the dendrify software framework[32] for accelerating the neural behavior simulation. However, today we still lack explicit and comprehensive studies on how to incorporate the temporal dendritic heterogeneity into a general SNN model and make it work in real-world temporal computing tasks, let alone explain how it works.

To solve the above challenges, we propose a novel LIF neuron model with temporal dendritic heterogeneity that also covers neural heterogeneity, termed DH-LIF. Then, we extend the neuron model to the network level, termed DH-SNNs, which support both the networks with only feedforward connections (DH-SFNNs) and those with recurrent connections (DH-SRNNs). We derive the explicit form of the learning method for DH-SNNs based on the emerging high-performance BPTT algorithm for ordinary SNNs[5,33]. By adaptively learning heterogeneous timing factors on different dendritic branches of the same neuron and on different neurons, DH-SNNs can generate multi-timescale temporal dynamics to capture features at different timescales. In order to reveal the underlying working mechanism, we elaborate a temporal spiking XOR problem and find that the interbranch feature integration in a neuron, the inter-neuron feature integration in a recurrent layer, and the inter-layer feature integration in a network have similar and synergetic effects in capturing multi-timescale temporal features. On extensive temporal computing benchmarks for speech recognition, visual recognition, EEG signal recognition, and robot place recognition, DH-SNNs achieve comprehensive performance benefits including the best reported accuracy along with promising robustness and generalization compared to ordinary SNNs. With an extra sparse restriction on dendritic connections, DH-SNNs present high model compactness and high execution efficiency on neuromorphic hardware. This work suggests that the temporal dendritic heterogeneity observed in the brain is a critical component in learning multi-timescale temporal dynamics, shedding light on a promising route for SNN modeling in performing complex temporal computing tasks.

## Results

### Spiking LIF neuron with temporal dendritic heterogeneity (DH-LIF)

Although neural network models have achieved tremendous success in practice, there is no doubt that a huge gap between the current neural network intelligence and the brain intelligence indeed exists, which motivates us to draw more inspiration from biology to improve the modeling. The brain presents many advantageous features while here we focus on the huge power in performing multi-timescale temporal computing tasks. As Fig. 1a depicts, the external stimuli such as languages and music injected into the brain usually present high temporal heterogeneity, i.e., showing variable timescales, but can be processed well by the brain. Furthermore, some biological recordings partially observed heterogeneous structures and multi-timescale dynamic responses across neurons and dendritic branches, which seems a link to the mentioned powerful functionality.

However, current neural network models do not sufficiently exploit the temporal heterogeneity in the brain, which might be a key reason that they cannot achieve satisfactory performance in performing multi-timescale temporal computing tasks. As presented in Fig. 1b, the artificial neuron in common ANNs simply models a linear summation of weighted synaptic inputs with a following nonlinear transfer function. This process without neural dynamics cannot model temporal memory. Note that, although ANNs can memorize temporal information by introducing recurrent connections to build recurrent neural networks (RNNs) and can further learn multi-timescale temporal dynamics by updating neural states asynchronously[34], the resulting extrinsic dynamics is different from the intrinsic dynamics within neurons discussed in this work and current RNNs do not model the dendritic heterogeneity that is just our focus. By contrast, the spiking neuron, for example, the classic simplified LIF neuron commonly used in ordinary SNNs, models temporal dynamics by updating the membrane potential of the soma over time with a decaying coefficient. In this work, we term the classic simplified LIF neuron and the decay coefficient of the membrane potential as the vanilla LIF neuron and the timing factor, respectively. The timing factor determines the timescale of neural responses, which consequently affects the spike rate. Moreover, it can be extended to achieve temporal neural heterogeneity by learning different timescales over different neurons[31]. However, it neglects the temporal heterogeneity on dendritic branches, which is widely observed in biological neurons[35]. This lack of temporal dendritic heterogeneity makes the simplified LIF neuron difficult to learn multi-timescale temporal information, thus failing to perform multi-timescale temporal computing tasks with high performance.

As illustrated in Fig. 1c, the main idea of this work lies in exploring how to improve SNNs for performing multi-timescale temporal computing tasks by incorporating temporal neural and dendritic heterogeneity. To this end, we abstract the cable properties of the dendrite and propose an enhanced LIF neuron model with temporal dendritic heterogeneity, termed DH-LIF (see Methods). Overall, a DH-LIF neuron is a multi-compartment model: a soma compartment with multiple dendrite compartments. As modeled in Fig. 2a, each dendritic branch has a temporal memory unit with a dendritic current variable $i_d$, which evolves like the membrane potential that is updated with pre-synaptic

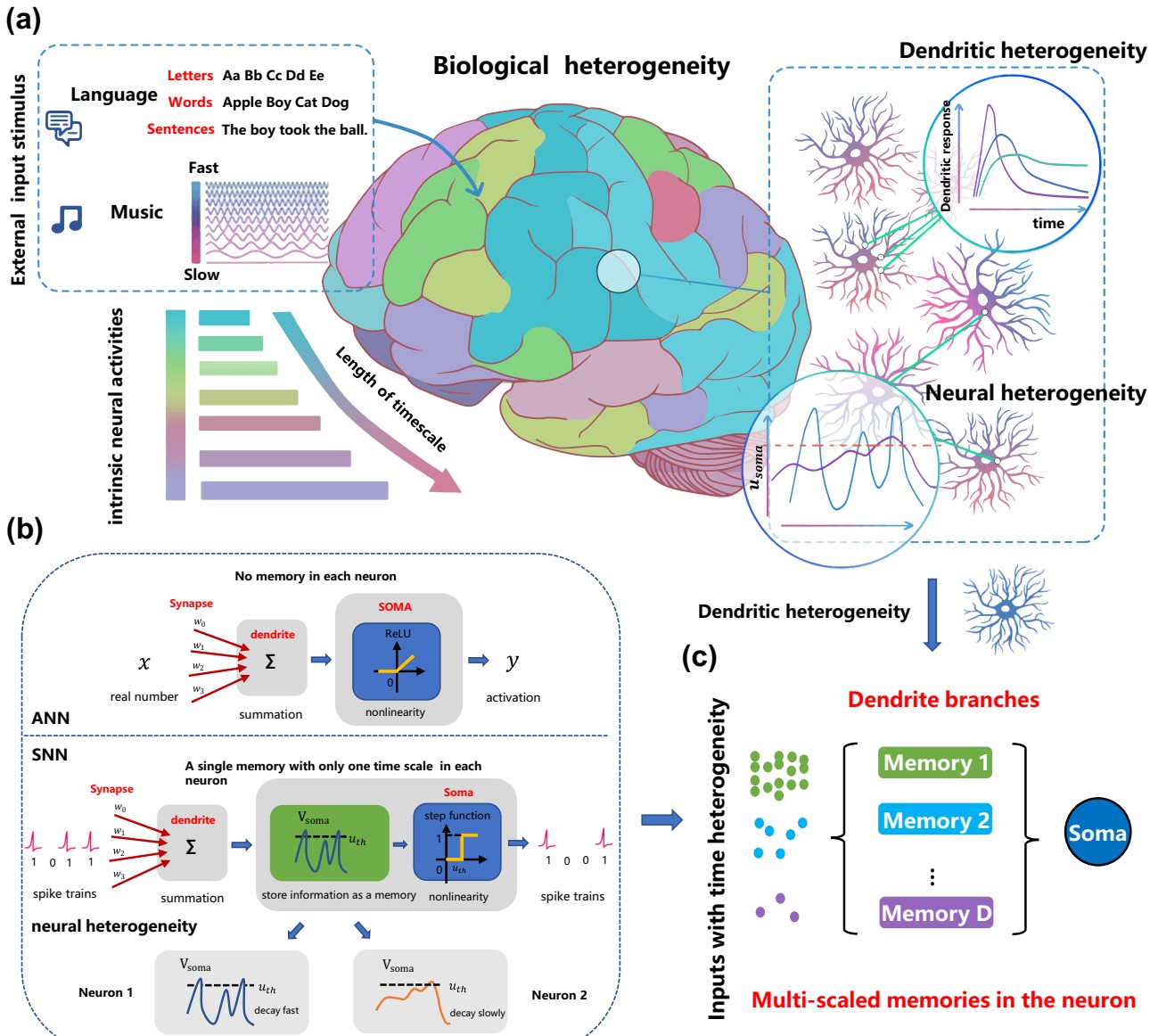

**Fig. 1 | Inspirations from biology to improve the modeling of SNNs with temporal dendritic heterogeneity. a** In the brain, there are rich timescales in the external stimuli and neural activities, and rich temporal heterogeneity in neural and dendritic responses[12,14]. **b** The artificial neuron model used in ANNs does not consider the temporal memory, while the spiking neuron model used in SNNs only considers a single-scale temporal memory in the neuronal membrane potential.

Existing SNNs can exhibit temporal neural heterogeneity by learning different timescales over different neurons but cannot memorize multi-timescale temporal information in a single neuron, which fail to perform complex temporal computing tasks with high performance. **c** This work aims at improving SNNs by incorporating temporal dendritic heterogeneity into the modeling for solving temporal computing tasks.

inputs and also decays by a timing factor, i.e., $\alpha_d$, every timestep. When different branches on a dendrite have different timing factors, the timescales of memorized information present temporal dendritic heterogeneity. Furthermore, the different timing factors of membrane potentialsddddd and dendritic currents in different neurons would also produce temporal neural heterogeneity.

Figure 2b provides an illustrative example that compares different responses between a DH-LIF neuron and a vanilla LIF neuron. Spike bursting is a common phenomenon observed in biological neurons. We assume that the neurons received two types of inputs: one is the high-frequency input that drives isolated spike events and the other is the low-frequency input that regulates the burst probability. This mechanism is similar to multiplexing, a known function of the dendrite[29]. In the illustration, we consider the illustration that DH-LIF neurons could generate bursting spikes while vanilla LIF neurons cannot. In the vanilla LIF neuron, there is only a soma memory unit

without dendritic memory. The timing factor of the membrane potential can only match the timescale of at most one of the two inputs, e.g., matching the high-frequency input (small timing factor) or matching the low-frequency input (large timing factor). When the neuron only matches the timescale of the high-frequency input, it loses long-term memory of the low-frequency input due to the fast decaying mechanism; when the neuron only matches the low-frequency input, it cannot closely track the high-frequency input due to the heavy memorization of historic information. Thus, as indicated in Fig. 2b, the vanilla LIF neuron cannot generate bursting spikes. In contrast, we can flexibly configure versatile timing factors on multiple dendritic branches in the DH-LIF neuron, which can make it capable of simultaneously dealing with variable timescales of different inputs, generating the bursting spikes successfully. In the prior work[29] about the multiplexing function of dendrites, the authors mentioned that they simulated the response of an ensemble of hick-tufted pyramidal neurons

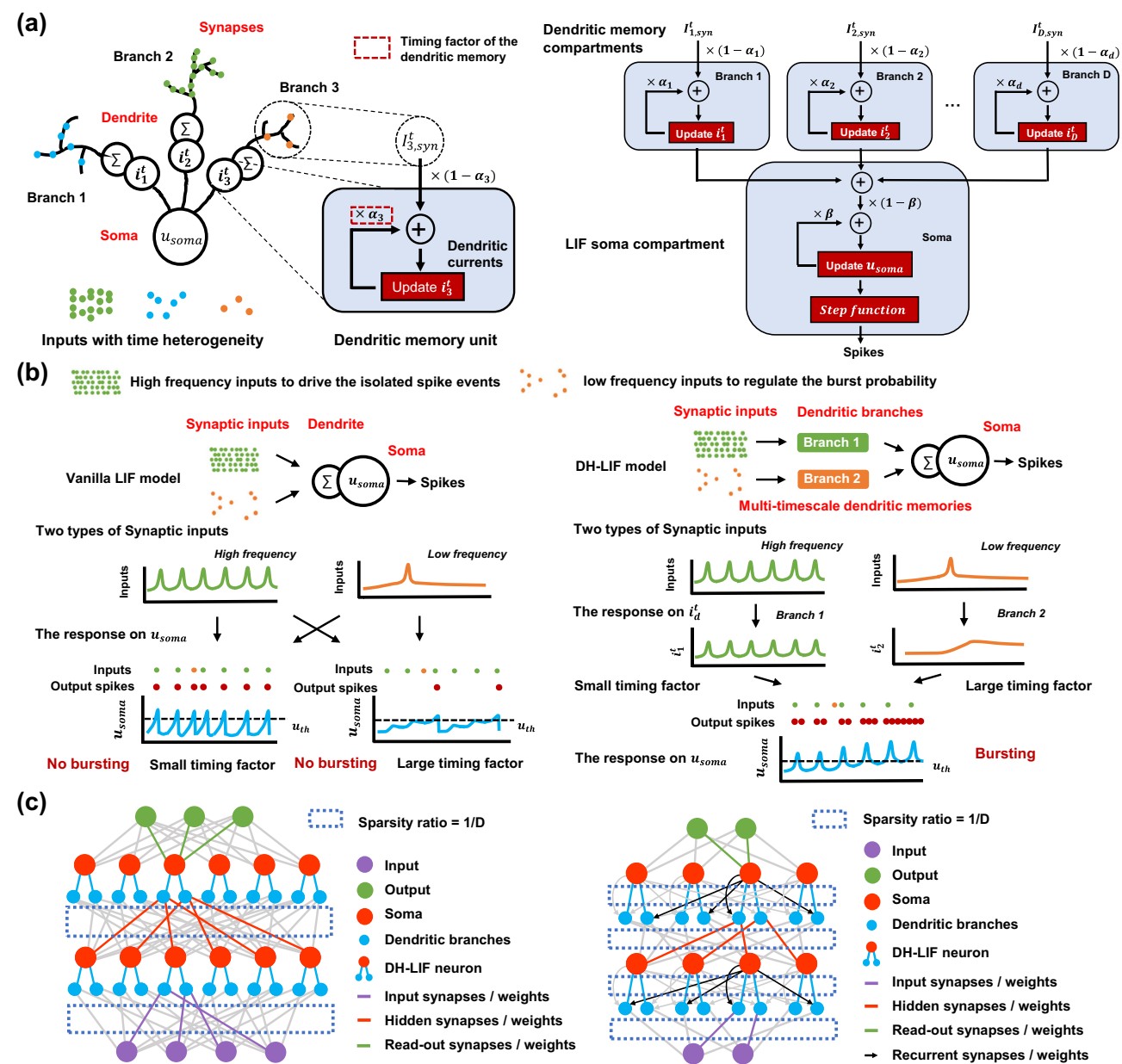

**Fig. 2 | The design of DH-LIF and DH-SNN. a** DH-LIF, a LIF neuron with temporal dendritic heterogeneity. DH-LIF is a multi-compartment neuron model with a soma compartment and multiple dendrite compartments. Besides the memory on the membrane potential of the soma, each dendritic branch has a temporal memory on the dendritic current with a variable timing factor highlighted in the red box. **b** Comparison between the responses of a vanilla LIF neuron and a DH-LIF neuron to mimic the phenomenon of bursting spikes. The DH-LIF neuron enables multi-timescale memories of information with different timescales. In this two-dendritic-branch example, the DH-LIF neuron presents both long-term memory of the low-frequency inputs that regulate the bursting probability and fast response to the high-frequency inputs that drive the isolated spike events, thus generating the bursting spikes successfully while the vanilla LIF neuron fails. **c** Illustration of DH-SFNN constructed by DH-LIF neurons with only feedforward connections and DH-SRNN with recurrent connections. The connections are sparse so that a DH-SNN model does not increase the number of parameters compared to vanilla LIF-based SNNs.

(TPNs) receiving two independent input signals with different frequencies: one injected into dendrites and the other injected into the soma. They further quantified the encoding quality in multiplexing at different timescales by calculating the frequency-resolved coherence between the inputs and the estimates. They found that the coherence between the dendritic inputs and the estimates based on the burst probability is close to one for slow input fluctuations, but decreases to zero for rapid input fluctuations, which is similar to our dendritic branch modeling with large timing factors. In the meantime, they found that the event rate can decode the soma input with high accuracy for input frequencies up to 100 Hz, which is similar to our dendritic branch modeling with small timing factors. Unlike the previous

focus on understanding hierarchical brain communication through multiplexing dendrites, we focus on the effectiveness of the proposed model inspired by biological observations for solving complex temporal computing tasks in practice with acceptable computational complexity and effective learning algorithms.

Modeling synaptic heterogeneity with a variable timing factor for each synapse can indeed offer valuable insights, but it comes with certain drawbacks, notably increased computation and storage overheads due to the large number of synapses (see Supplementary Table S4). As the number of dendritic branches increases, the dendritic heterogeneity can provide a reasonable approximation of the synaptic heterogeneity. However, the experimental results in Supplementary

Fig. S6 imply that an overlarge number of dendritic branches may saturate or even negatively impact the performance. Therefore, for solving real-world temporal computing tasks, it is wiser to incorporate the dendritic heterogeneity rather than the synaptic heterogeneity. Doing so would create a better balance between computational efficiency and the ability to model the expected multi-timescale dynamics. Generally speaking, a DH-LIF neuron has both long-term memory of the low-frequency input and fast response to the high-frequency input with reasonable computation and storage overhead, which simultaneously reflects the rich temporal heterogeneity and promises practical use in solving real-world temporal computing tasks.

### Spiking neural network with DH-LIF neurons (DH-SNN)

Based on the proposed DH-LIF neuron model, we further construct SNNs with temporal dendritic heterogeneity, termed DH-SNN. Specially, the DH-SNN with only feedforward connections is denoted as DH-SFNN while the one with recurrent connections is denoted as DH-SRNN, as illustrated in Fig. 2c. In order to avoid the parameter exploding as the number of dendritic branches grows, we add a sparse restriction on the connection pattern between neurons (see Methods). For each neuron, the pre-synaptic inputs are randomly distributed on the dendritic branches. The sets of input indexes on different branches are non-overlapped and the number of inputs keeps identical across branches to the greatest extent. The connection restriction is valid for both DH-SFNNs and DH-SRNNs, which guarantees a constant parameter volume when the number of dendritic branches grows and makes the number of parameters comparable to SNNs constructed with vanilla LIF neurons, termed as vanilla SNNs. This is important for saving storage and computational costs when deploying the model on hardware for efficient execution, and can also reflect the fact that our performance improvements are indeed benefited from the introduced temporal dendritic heterogeneity rather than using more parameters. Our following experiments will provide explanations on the working mechanism of DH-SFNNs and DH-SRNNs in performing multi-timescale temporal computing tasks.

For a network with many timing factors, it is difficult to manually configure their values for achieving optimal application performance. In order to gain high performance in practical tasks, automatic learning of timing factors to shape the landscape of temporal heterogeneity is highly expected. We adapt the emerging SNN-version BPTT learning algorithm for DH-SNNs to explicitly calculate gradients (see Methods), which also allows convenient comparison with state-of-the-art baselines using similar learning algorithms. In our framework, synaptic weights, timing factors of membrane potentials, and timing factors of dendritic currents are all learned automatically during the training phase. When all dendritic timing factors are small enough, the dendrites would lose the memorization capability thus degrading to vanilla SNNs without dendritic heterogeneity. Therefore, it is intuitive that DH-SNNs can perform better than vanilla SNNs since the latter is just a special case of the former, which would be supported by the following experimental results.

### Long-term memory via dendritic dynamics

The temporal dynamics in each neuron endows SNNs with the capability to memorize historic information. In a vanilla LIF neuron, the membrane potential, i.e., $u$, can be viewed as the memory of historic information. Long-term memory can be achieved by configuring a large timing factor on the membrane potential, i.e., $\beta$, for slowing the membrane potential decaying. However, we argue that a vanilla LIF neuron cannot truly memorize information for a long time even with a large $\beta$ value due to the reset mechanism of the membrane potential every time the neuron fires a spike. Fortunately, the proposed DH-LIF succeeds in preserving long-term information owing to the multi-compartment modeling. Although the soma suffers the reset mechanism, the dendritic current on each dendritic branch will never

be reset. In this way, the temporal dendritic dynamics enables long-term memory.

To evidence our prediction, we design a delayed spiking XOR problem for testing the capability of long-term memory of vanilla SFNNs and DH-SFNNs. For simplicity, we assign only one dendritic branch for each neuron in DH-SFNNs. As illustrated in Fig. 3a, the delayed spiking XOR problem experiences three stages. In the first stage, an initial spike pattern with a low or high firing rate is injected into the network. In the second stage, the model goes through a long delay duration with some noisy spikes. Last, the model receives another input spike pattern and outputs the result of the XOR problem (the ground truths are denoted as labels) by conducting an XOR operation between the initial and final input spike patterns. Specifically, the output result considers the firing rates of the input spike patterns at the beginning and the end, behaving like an XOR operation as the right truth table shows. The network structures can be found in Supplementary Fig. S1. With the delayed spiking XOR problem, we can easily test the memory capability of the models by configuring different delay values. Notice that here the DH-LIF neuron with one dendritic branch is similar to an existing model[31] in which the dendritic current is called synaptic current. However, that work focused on neural heterogeneity across neurons rather than the dendritic heterogeneity across both dendritic branches and neurons in our work. Although similar experiments can be conducted with the existing model, the role of dendritic dynamics was not explicitly analyzed.

The experimental results are depicted in Fig. 3b. Testing models include vanilla SFNNs and one-dendritic-branch DH-SFNNs with different initial distributions of timing factors. Notice that unless otherwise specified, the timing factors of membrane potentials, $\beta$, are initialized following a medium distribution and are learnable in the following experiments of Fig. 3. More analyses on the initialization and learning of membrane potential timing factors can be found in Supplementary Fig. S2. The dendritic timing factors, $\alpha$, can be fixed or learnable during training. It turns out that one-dendritic-branch DH-SFNNs significantly outperform vanilla SFNNs in the delayed spiking XOR problem, showing longer-term memory. This conclusion holds no matter whether the dendritic timing factors in DH-SFNNs are fixed or learnable, which reflects the good preservation of historic information in dendritic currents without the reset mechanism. Figure 3c further presents gradients of the loss with respect to membrane potentials of the vanilla SFNN and to dendritic currents of the DH-SFNN through time at the beginning of training under large initialized timing factors. The gradients with respect to membrane potentials in the vanilla SFNN quickly vanish after backpropagating a period of time even though given large timing factors, while the gradients with respect to the dendritic currents can hold for a long time. This difference is caused by the reset mechanism of the membrane potential that cleans the memorized historic information, which is further analyzed in Supplementary Fig. S3 and discussed in Methods. Two more conclusions can be observed: (1) Larger initialized timing factors produce longer-term memory than smaller ones owing to the slow decaying of historic information; (2) Learnable dendritic timing factors produce longer-term memory than fixed ones. The accuracy can be greatly improved especially when the initialized timing factors are inappropriate for the task, i.e., smaller distributions here. From Fig. 3d, it can be seen that the model training drives some small and medium initialized dendritic timing factors to larger values for maintaining longer length of memory.

Besides the delayed spiking XOR problem, we extend our comparison on speech benchmarks, i.e., SHD and SSC datasets. The spike patterns of the two datasets are visualized in Fig. 3e and more temporal characterizations are provided in Supplementary Fig. S5, which demonstrate the rich timescales of the two datasets and implies the need for temporal heterogeneity in the processing model. As demonstrated in Fig. 3f, one-dendritic-branch DH-SFNNs with

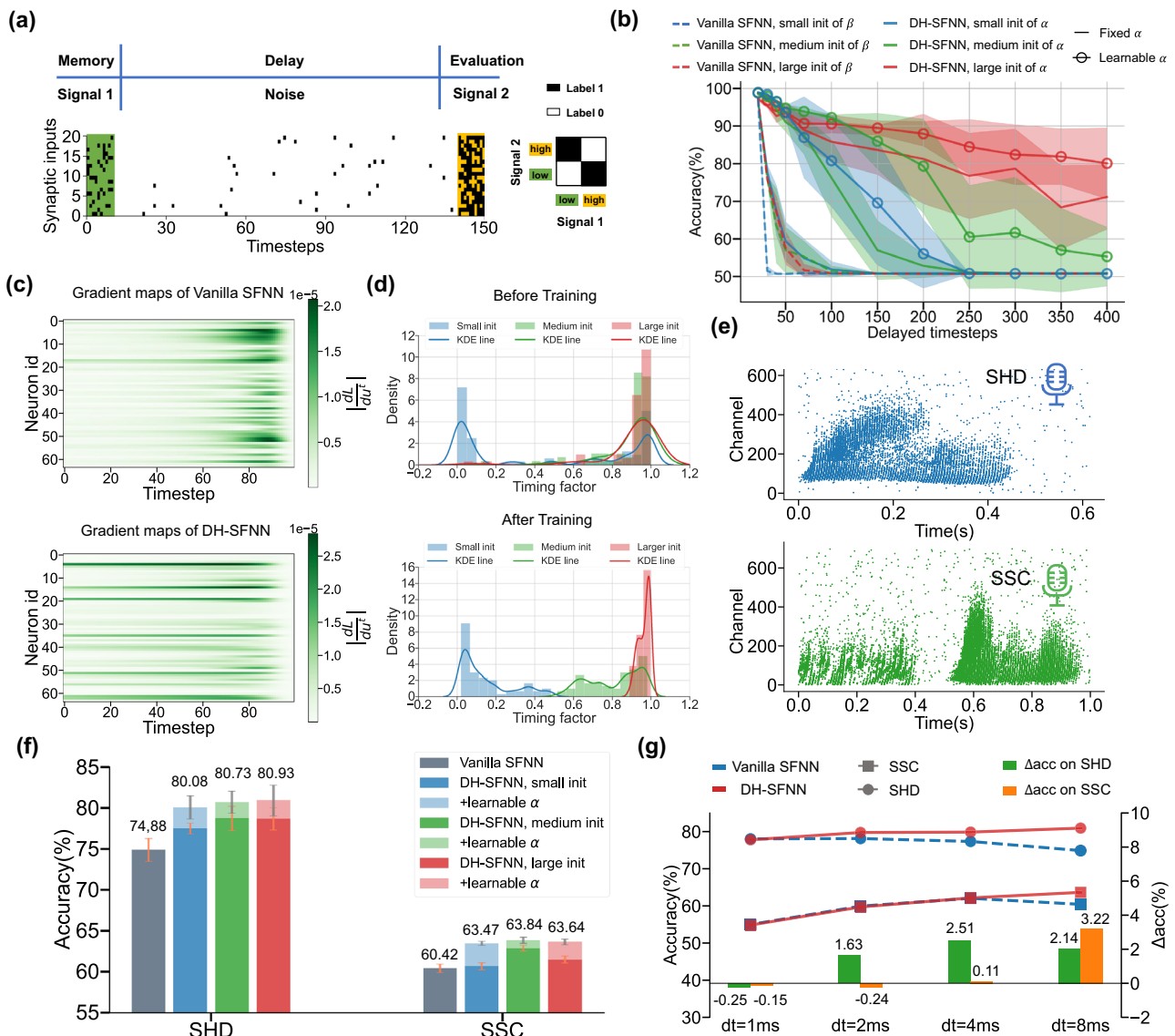

**Fig. 3 | Long-term memory on dendrites. a** Illustration of the delayed spiking XOR problem for testing the memory length of vanilla SFNNs and DH-SFNNs with only one dendritic branch in each DH-LIF neuron. **b** Accuracy curves of vanilla SFNNs and one-dendritic-branch DH-SFNNs. For vanilla SNNs, the timing factors of membrane potentials, $\beta$, are learnable with three different initialized distributions. For DH-SNNs, $\beta$ are learnable and initialized with a medium distribution, and the dendritic timing factors, $\alpha$, can be fixed or learnable with three different initialized distributions. **c** Visualizing gradients of the loss with respect to membrane potentials of the vanilla SFNN and to dendritic currents of the one-dendritic-branch DH-SFNN through time at the beginning of training under large initialized timing factors. **d** Distributions of dendritic timing factors before and after training. KDE line, kernel density estimate line. **e** Examples of the input spike trains from SHD and SSC datasets. **f** Comparing recognition accuracy of vanilla SFNNs and one-dendritic-branch DH-SFNNs with fixed or learnable dendritic timing factors on SHD and SSC under the sampling time interval of $dt = 1\,ms$. **g** Comparing recognition accuracy of vanilla SFNNs and one-dendritic-branch DH-SFNNs with learnable timing factors under different sampling time intervals on SHD and SSC. A beneficial initialization of timing factors is selected for each sampling time interval to demonstrate overall better accuracy. In above experiments, unless otherwise specified, the timing factors of membrane potentials are initialized following a medium distribution and are learnable during training. The standard deviations (presented as error bars) represent 10 or 5 repeated trials for the spiking XOR problem or other tasks, respectively.

learnable dendritic timing factors achieve much better accuracy than vanilla SFNNs no matter the initialized distributions. Under the sampling time interval $dt = 1\,ms$, our one-dendritic-branch DH-SFNNs with large initialized timing factors achieve 82.2% accuracy on SHD and 63.62% accuracy on SSC, which are 4.65% and 5.16% higher than vanilla SFNNs on SHD and SSC, respectively. To further support our claims, we change the timescale of the input spike patterns by tuning the sampling time interval from $dt = 1\,ms$ to $dt = 8\,ms$. A smaller $dt$ implies better sampling precision, a slower timescale, and a longer time window. Notice that here we choose appropriate initialized timing factors for each $dt$ setting to show overall better accuracy, i.e., initializing larger timing factors for smaller $dt$ values. As presented in Fig. 3g, the

accuracy of vanilla SFNNs do not always improve and even degrades as the sampling precision grows. The accuracy gap between DH-SFNNs and vanilla SFNNs tends to increase as $dt$ decreases because DH-SFNNs do better in long-term memory.

## Intra-neuron heterogeneous feature integration

via multi-branch dendrites. In the above section, we have demonstrated the long-term memory of DH-SNNs benefited from the temporal dynamics on the dendritic branch. We have predicted in Fig. 2 that DH-SFNNs with multiple branches in each neuron can perform temporal computing tasks via temporal heterogeneity. In this section, we upgrade the above delayed spiking XOR problem to a multi-

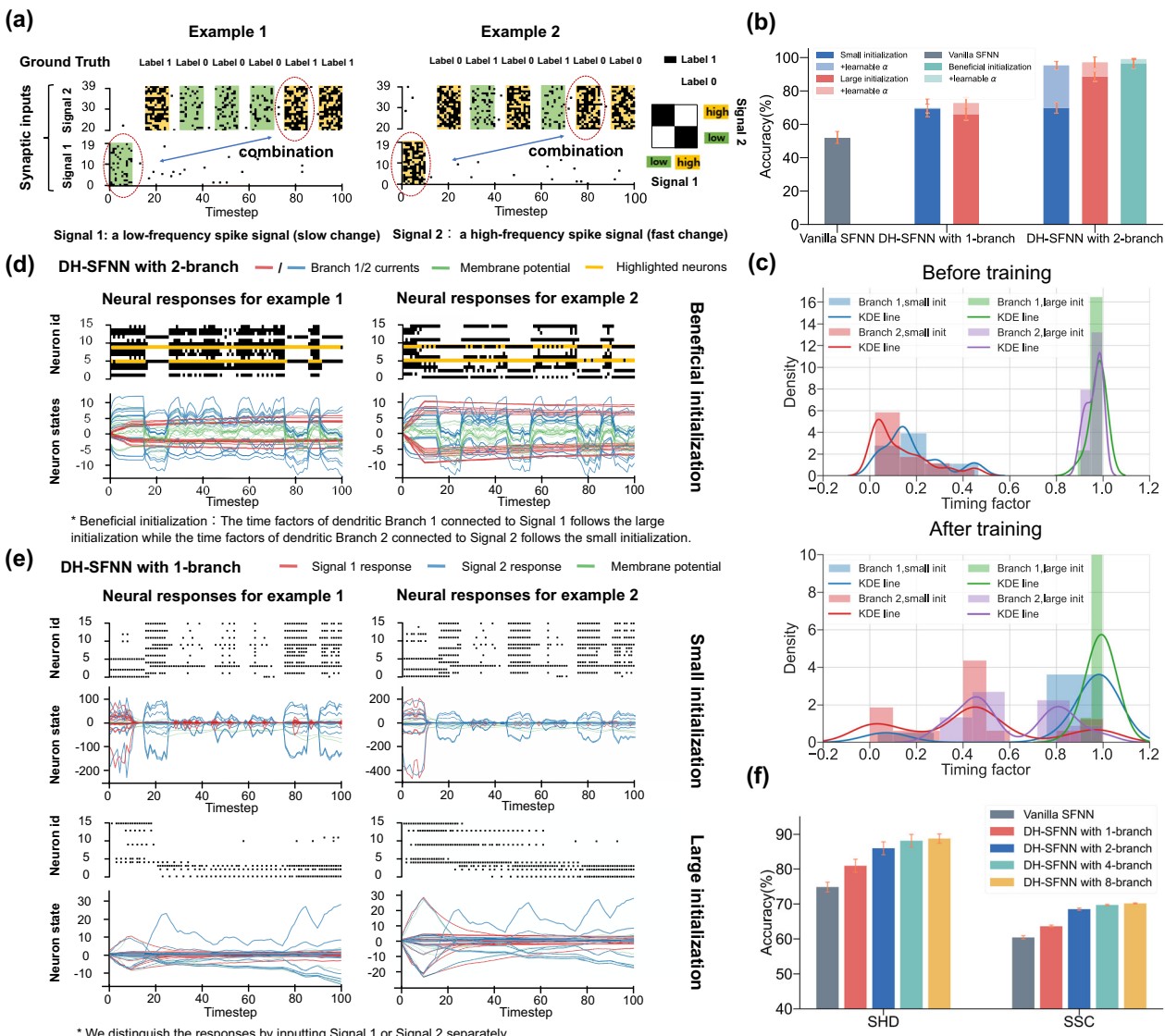

**Fig. 4 | Intra-neuron heterogeneous feature integration through multiple dendritic branches. a** Illustration of the multi-timescale spiking XOR problem for testing the capability of processing temporally heterogeneous information of the vanilla SFNN and DH-SFNNs with one or two dendritic branches in each DH-LIF neuron. **b** Comparing accuracy of the vanilla SFNN and DH-SFNNs with different numbers of dendritic branches and different initial distributions of timing factors. The dendritic timing factors, $\boldsymbol{\alpha}$, can be fixed or learnable during training. The beneficial initialization means that we initialize large dendritic timing factors for Branch 1 in each DH-LIF neuron while initializing small dendritic timing factors for Branch 2. **c** Distributions of dendritic timing factors of two dendritic branches before and after training. KDE line, kernel density estimate line. **d** Visualization of the output spike pattern and dendritic currents of two-dendritic-branch DH-LIF neurons with fixed timing factors during training under a beneficial initialization. **e** Visualization of the output spike pattern and dendritic currents of one-dendritic-branch DH-LIF neurons with fixed timing factors during training under a small or large initialization. **f** Comparing recognition accuracy of vanilla SFNNs and DH-SFNNs with variable numbers of dendritic branches and learnable timing factors under a large distribution on SHD and SSC. The sampling time interval is set to $dt=1ms$. In above experiments, unless otherwise specified, the timing factors of membrane potentials are initialized following a medium distribution and are learnable during training. The standard deviations (presented as error bars) represent 10 or 5 repeated trials for the spiking XOR problem or other tasks, respectively.

timescale spiking XOR problem for testing the model's capability of processing temporally heterogeneous information to further support our prediction. As depicted in Fig. 4a, the multi-timescale spiking XOR problem uses two types of input spike signals. At the first stage, a single spike pattern (Signal 1) with a low (left) or high (right) firing rate is fed into the model, representing a low-frequency component. Then, several similar spike patterns with faster periods (Signal 2) are injected into the model sequentially, representing a high-frequency component. Each time the model receives a spike pattern in Signal 2, it also outputs an XOR result between the beginning spike pattern in Signal 1 and the current spike pattern in Signal 2. The goal of the model is to memorize the low-frequency Signal 1 and conduct an XOR operation

with the high-frequency Signal 2, which can substantially reflect its potential capability of processing temporally heterogeneous information.

In the multi-timescale spiking XOR problem, we compare the vanilla SFNN and DH-SFNNs with one or two dendritic branches and only one hidden layer. For the two-dendritic-branch DH-LIF neuron, the input synapses carrying Signal 1 are connected to one branch (Branch 1) while the synapses carrying Signal 2 are connected to the other branch (Branch 2). The network structure can be found in Supplementary Fig. S1. Figure 4b presents the accuracy results. The vanilla SFNN fails in performing this task with an accuracy much lower than 75%. Although one-dendritic-branch DH-SFNNs have long-term

memory as evidenced by Fig. 3, they cannot process information with multiple timescales well. As the number of branches grows to two, DH-SFNNs demonstrate much better performance owing to the temporal dendritic heterogeneity especially when the dendritic timing factors are initialized appropriately and learnable. Here the beneficial initialization means that we initialize large dendritic timing factors for Branch 1 in each DH-LIF neuron to enable long-term memory for low-frequency Signal 1 while initializing small dendritic timing factors for Branch 2 to enable fast response for high-frequency Signal 2. Figure 4c visualizes the dendritic timing factors before and after training. As expected, the dendritic timing factors of Branch 1 with a small initialization tend to become larger while the dendritic timing factors of Branch 2 with a large initialization tend to become smaller, which evidences that the learning process makes the dendritic timing factors match the multiple timescales of input signals better. Notice that unless otherwise specified, the timing factors of membrane potentials are initialized following a medium distribution and are learnable in the experiments of Fig. 4.

In Fig. 4d, we further visualize the output spike pattern and dendritic currents of two-dendritic-branch DH-LIF neurons with fixed timing factors during training under a beneficial initialization. The left and right results correspond to the left and right input cases in Fig. 4a, respectively. With large dendritic timing factors on Branch 1, the low-frequency Signal 1 can be memorized for a long term by the dendritic currents on Branch 1; meanwhile, with small dendritic timing factors on Branch 2, the high-frequency Signal 2 can be closely tracked by the dendritic currents on Branch 2. The dendritic currents on two branches with different timescales are integrated to synergistically determine the membrane potentials and output spikes. Interestingly, after learning synaptic weights, some neurons learn features that are sensitive to reflect a specific combination of Signal 1 and Signal 2. For example, we highlight two neurons in Fig. 4d, whose spikes are retained in black while the areas with no spikes are marked in yellow. The first highlighted DH-LIF neuron (i.e., with a smaller neuron ID) is sensitive to the case of combining low-firing-rate Signal 1 and high-firing-rate Signal 2, while the second highlighted DH-LIF neuron (i.e., with a larger neuron ID) is sensitive to the case of combining high-firing-rate Signal 1 and high-firing-rate Signal 2. Here 'sensitive' means firing consecutive spikes corresponding to the learned combination feature between Signal 1 and Signal 2. These specific learned features of DH-LIF neurons are critical for performing the multi-timescale spiking XOR problem correctly in the following decision layer. For comparison, we make the similar visualization for one-dendritic-branch DH-LIF neurons in Fig. 4e. If we initialize small dendritic timing factors, the dendritic currents cannot memorize the low-frequency Signal 1 and are mainly controlled by the high-frequency Signal 2. For large initialized dendritic timing factors, the dendritic currents cannot tightly track the high-frequency Signal 2. Therefore, the one-dendritic-branch DH-SFNN, as well as the SNNs with synaptic current dynamics[31] equivalent to our one-dendritic-branch DH-SNNs, cannot learn specific features combining Signal 1 and Signal 2, failing in performing the multi-timescale spiking XOR problem. In addition, we conduct extra experiments in which Signal 1 and Signal 2 are randomly connected to the two dendritic branches of DH-SFNNs without connection restriction. The two-dendritic-branch DH-SFNNs succeed in handling the problem indicating the connection restriction is unnecessary and DH-SNNs can still acquire selectivity to multiple timescales of input signals during the learning process (see Supplementary Fig. S4). Our experimental results explain that multiple dendritic branches of a neuron are able to simultaneously match different timescales, which allows DH-SNNs to make complex decisions in the temporal domain via feature integration thus enhancing the capability of performing multi-timescale temporal computing tasks.

Besides the synthetic multi-timescale spiking XOR problem, we also compare the performance of vanilla SFNNs, one-dendritic-branch DH-SFNNs, and DH-SFNNs with two or more dendritic branches on SHD and SSC datasets. We keep the same network architecture as that used in Fig. 3f. Because the timescales of SHD and SSC are more complicated than the two timescales in the above XOR problem, it is hard to find the beneficial initialization. For simplicity, we initialize all dendritic timing factors following a large initialization and make them learnable to automatically match different timescales. As Fig. 4(f) depicts, DH-SFNNs with more dendritic branches achieve better recognition accuracy on both datasets. Although the improvement tends to be saturated with redundant dendritic branches (see more analyses in Supplementary Fig. S6), the results indeed prove that the temporal dendritic heterogeneity of DH-LIF neurons can enhance the representation power of DH-SFNNs for performing multi-timescale temporal computing tasks.

### Inter-neuron feature integration via synaptic connections

In the above experiments, we have revealed that the temporal features with different timescales can be integrated by multiple dendritic branches within DH-LIF neurons. In this subsection, we try to demonstrate another route for integrating multi-timescale temporal features through synaptic connections. Here synaptic connections include feedforward connections between layers in SFNNs and recurrent connections within layers in SRNNs.

To support our prediction, we test the one-layer DH-SFNN, the two-layer DH-SFNN, and the one-layer DH-SRNN, all of which only have one dendritic branch in each DH-LIF neuron to eliminate the influence of intra-neuron feature integration. The task is the same multi-timescale spiking XOR problem as above and the network structures can be found in Supplementary Fig. S1. We find that the one-layer DH-SFNN fails in performing this task, while both the two-layer DH-SFNN and the one-layer DH-SRNN perform well with about 99% accuracy. In Fig. 5a, we show the output spike pattern of each layer in the models. Specially, we find three types of neurons. The Type 1 neuron represents the neuron sensitive to Signal 2 with high frequency and the Type 2 neuron represents the neuron sensitive to Signal 1 with low frequency. Notice that we identify Type 2 neurons by comparing the same neurons' responses in different input cases as the left and right panels present. For example, looking at the first hidden layer of the two-layer SFNNs, when Signal 1 with a low firing rate is inputted in the left panel, Type 2 neurons exhibit sparse spiking activities. On the contrary, when Signal 1 with a high firing rate is inputted in the right panel, Type 2 neurons display dense spiking activities. Furthermore, the responses of Type 2 neurons are uniformly distributed, not influenced by the periodically changing Signal 2. With these observations, we conclude that Type 2 neurons are sensitive to Signal 1. The highlighted neuron represents the neuron sensitive to a specific combination of Signal 1 and Signal 2 which is critical for the correct functionality of solving the multi-timescale spiking XOR problem. For the two-layer DH-SFNN, we find that there are only Type 1 and Type 2 neurons in the first hidden layer because a DH-LIF neuron with only one dendritic branch can only capture single-scale temporal features. In the second hidden layer, highlighted neurons come out by integrating the output spike patterns of Type 1 and Type 2 neurons. For example, the two highlighted neurons here are sensitive to the case of combining low-firing-rate Signal 1 and high-firing-rate Signal 2. While for the one-layer DH-SRNN, Type 2 and highlighted neurons are observed in the first hidden layer. Similarly, the two highlighted neurons here are sensitive to the case of combining high-firing-rate Signal 1 and high-firing-rate Signal 2. In the DH-SRNN, highlighted neurons can access features of Signal 1 memorized by Type 2 neurons through the recurrent synaptic connections. Particularly, the high-frequency Signal 2 features are received instantaneously and can be combined with the memorized Signal 1 features to activate highlighted neurons. This experiment visually evidences

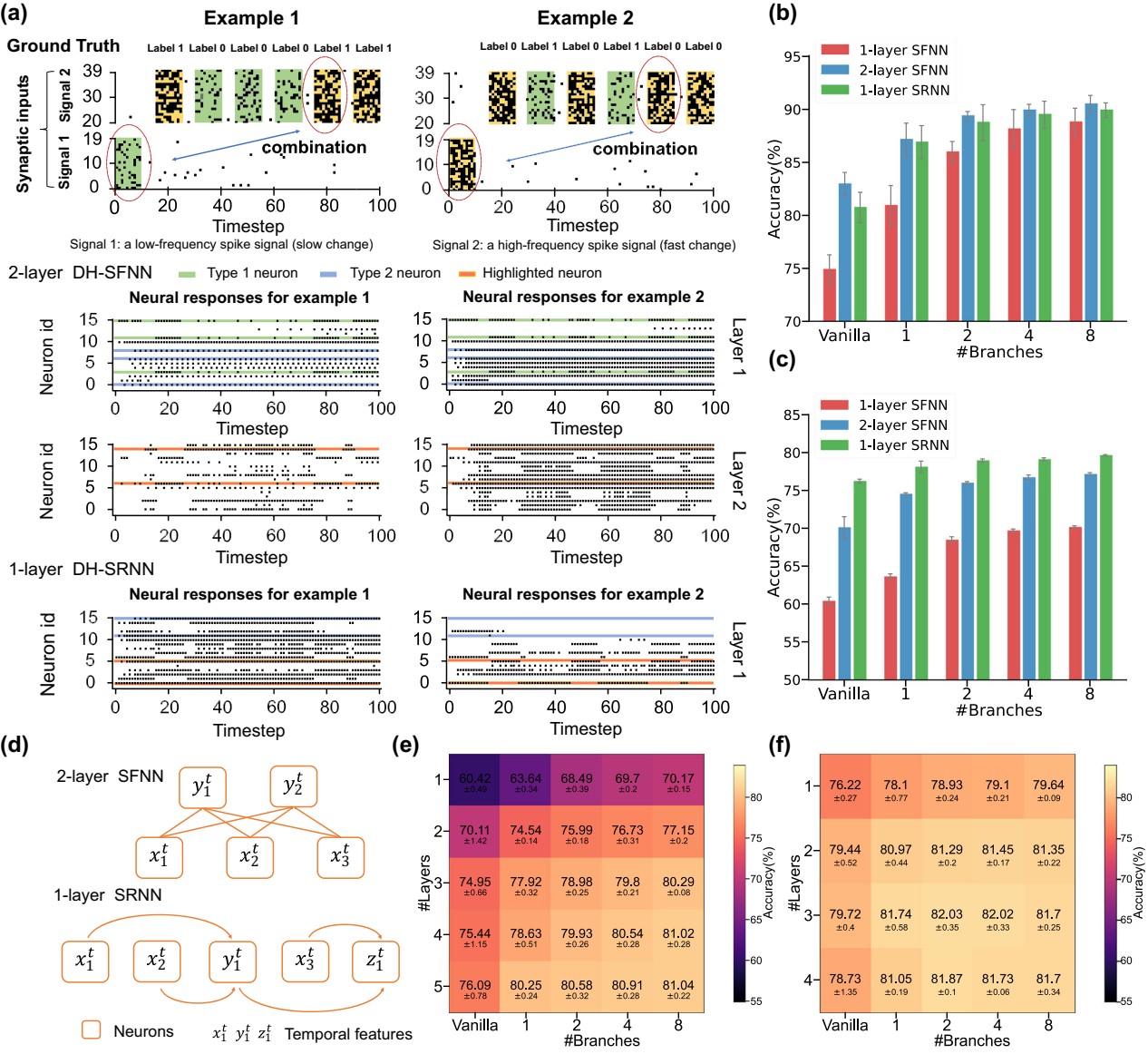

**Fig. 5 | Inter-neuron heterogeneous feature integration through synaptic connections. a** Visualization of output spike patterns of the two-layer DH-SFNN (middle) and the one-layer DH-SRNN (bottom) with one dendritic branch in each DH-LIF neuron when performing the multi-timescale spiking XOR problem (top). **b** Comparing accuracy of one-layer or two-layer SFNNs and DH-SFNNs, and one-layer SRNNs and DH-SRNNs on the SHD dataset. **c** Comparing accuracy of one-layer or two-layer SFNNs and DH-SFNNs, and one-layer SRNNs and DH-SRNNs on the SSC dataset. **d** Illustration of the feedforward and recurrent synaptic connections for performance analysis. **e** Accuracy results of multi-layer SFNNs and DH-SFNNs with different numbers of dendritic branches on the SSC dataset. **f** Accuracy results of multi-layer SRNNs and DH-SRNNs with different numbers of dendritic branches on the SSC dataset. In the above experiments, unless otherwise specified, the number of layers represents the number of hidden layers and the timing factors of membrane potentials are initialized following a medium distribution and are learnable during training. In the experiments on SHD and SSC datasets, we test learnable dendritic timing factors under both medium and large initializations and present the best results. The standard deviations represent 5 repeated trials.

the integration of multi-timescale temporal features via synaptic connections in feedforward and recurrent networks, which helps perform multi-timescale temporal computing tasks.

Given the above analyses, it looks clear that the inter-branch feature integration in a neuron, the inter-neuron feature integration in a recurrent layer, and the inter-layer feature integration in a network have similar and synergetic effects in capturing the multi-timescale temporal features, which are beneficial for performing multi-timescale temporal computing tasks. To provide more evidence, we conduct extra experiments with a variable number of dendritic branches on SHD and SSC datasets. The results are presented in Fig. 5b, c, from which several conclusions can be drawn. First, DH-LIF neurons improve

the capability of handling temporal heterogeneity with higher accuracy. Second, compared to one-layer SFNNs, two-layer SFNNs and one-layer SRNNs demonstrate much better performance owing to the inter-neuron integration of temporal features. Third, DH-SFNNs and DH-SRNNs gradually produce higher accuracy as the number of dendritic branches grows. In short, these results evidence the improved capability of performing multi-timescale temporal computing tasks benefited from the temporal dendritic heterogeneity, and further reveal the synergistic working mechanism of the neuron-level and network-level feature integration.

Specifically, we observe that one-layer SRNNs tend to perform better than two-layer SFNNs, especially on the SSC dataset with higher

difficulty. For analysis, we illustrate the connection topology of a two-layer SFNN and a one-layer SRNN in Fig. 5d as an example. Apparently, a neuron in the second hidden layer of a two-layer SFNN can only spatially integrate the learned features of the previous layer once to form a slightly higher-level feature. In contrast, the recurrent connections can help neurons in a one-layer SRNN integrate the learned features multiple times to form much higher-level features. For example, the low-level features $x_1^t$ and $x_2^t$ are integrated together to generate a slightly higher-level feature $y_1^t$, while $y_1^t$ is further integrated with $x_3^t$ to generate a much higher-level feature $z_1^t$. Furthermore, we compare two-layer SFNNs and one-layer SRNNs with wider one-layer SFNNs (see Supplementary Fig. S7). Here 'wider' means more neurons in the hidden layer, thus leading to more parameters. The results show that the performance improvement of wider one-layer SFNNs is not as significant as that by introducing inter-neuron feature integration in two-layer SFNNs and one-layer SRNNs, which implies that the performance improvement in performing multi-timescale temporal computing tasks cannot be simply achieved by increasing the number of parameters.

Beyond two-layer SFNNs and one-layer SRNNs for basic analysis, we further design experiments with multi-layer SFNNs and SRNNs on the SSC dataset. The reason for selecting SSC is because it is more complicated than SHD, which can provide a wider exploration space of model performance. The results are depicted in Fig. 5e, f). Generally, the accuracy scores of SFNNs and SRNNs tend to increase as the number of layers grows. Similar trends are also observed as the number of dendritic branches grows. Meanwhile, we find that the accuracy gap between SFNNs and SRNNs is narrowed as the number of layers or dendritic branches increases, which indicates that the performance of different models will become saturated when the integration extent of temporal features is enough for the model to perform the task. Specifically, in deeper layers, the accuracy saturation appears when increasing the number of dendritic branches, and this trend in SRNNs with more comprehensive feature integration can be faster than that in SFNNs. Notice that the complexity introduced by the increasing number of layers makes deeper models sometimes challenging to train, which might also degrade the model performance.

## Comprehensive performance benefits of DH-SNNs

Usually, a DH-LIF neuron has more parameters than a vanilla LIF neuron. At the neuron level, there are additional timing factors on dendritic branches, whose volume is proportional to the number of dendritic branches. At the network level, the number of synapses would explode if each dendritic branch is connected to all synaptic inputs. To reduce the parameter volume, we add a sparse restriction on the synaptic connection pattern, i.e., each dendritic branch only connects to a part of synaptic inputs and the number of synapses on each dendritic branch is balanced to a great extent (see Methods). In this way, DH-SNNs do not obviously increase storage and computational costs compared to vanilla SNNs. As given in Fig. 6a, the increase of parameters of DH-SNNs over vanilla SNNs is neuron-wise and proportional to the number of dendritic branches, which can be neglected compared to the heavy synaptic weights. Furthermore, we quantitatively present the numbers of parameters and synaptic operations of vanilla SNNs and DH-SNNs with different numbers of dendritic branches. We collect results from one-layer SFNNs and SRNNs on the SSC dataset and show the results in Fig. 6b. As predicted, there is no obvious increase of parameters and synaptic operations as the number of dendritic branches grows. The occasional fluctuation of synaptic operations is caused by the variable firing rate in different models. We further test our models on extensive datasets, including two speech datasets (GSC[36] and TIMIT[37]), two spiking speech datasets (SHD and SSC)[38], and two sequence datasets (S-MNIST and PS-MNIST). The experimental settings can be found in Methods and results are provided in Table 1. On these datasets, we find that our proposed DH-SNNs

improve accuracy significantly over other SNNs and long short-term memory (LSTM) models even using much fewer parameters. In particular, on SHD, compared to the best reported accuracy of SNNs[33], our models can improve accuracy from 90.4% to 92.1% with only 36% parameters; on SSC, our models boost the best reported accuracy from 74.2% to 82.46% with only 45% parameters. Supplementary Table S5 also shows that our DH-SNNs enjoy much higher computational efficiency over LSTM models on these two datasets, as high as hundreds to thousands of times. On the classic benchmarks commonly used for speech recognition tasks, i.e., GSC and TIMIT with non-spiking data, our DH-SNNs with much fewer parameters again obtain better accuracy compared to previous SNN models. On datasets with less temporal heterogeneity such as S-MNIST and PS-MNIST, DH-SNNs also demonstrate competitive accuracy.

The robustness of SNNs can also be enhanced by temporal dendritic heterogeneity. We add random spike noises into the original data for testing the robustness of vanilla SFNNs and DH-SFNNs in resisting noises. The random spike noises follow a Poisson distribution with variable rates. As depicted in Fig. 6c, DH-SFNNs with multiple dendritic branches suffer from slower accuracy degradation as the noise rate increases, thus presenting better robustness. For vanilla SFNNs without dendritic modeling or DH-SFNNs with only one dendritic branch, all synaptic inputs are directly concentrated at the soma or on the only dendritic branch. Therefore, each noisy input would influence the entire dynamics of the neuron. When the timing factor of the membrane potential or the dendritic current is large, the disturbance caused by the noise decays slowly and accumulates gradually, finally harming model performance. Fortunately, for DH-SFNNs with multiple dendritic branches, synaptic inputs are distributed on different dendritic branches. Owing to the rich temporal dendritic heterogeneity, there is usually a part of dendritic timing factors being small, which would decay the disturbance caused by noises on those dendritic branches fast, greatly reducing the influence on the entire dynamics of the neuron. In this way, DH-SFNNs with multiple dendritic branches enjoy better robustness than vanilla SFNNs. We also observe similar results on vanilla SRNNs and DH-SRNNs (see Supplementary Fig. S8). Besides robustness, we further test the generalization capability by pre-training models under a sampling time interval and fine-tuning them under a different time interval (see Supplementary Fig. S9). Again owing to the natural temporal heterogeneity, DH-SNNs with multiple dendritic branches demonstrate better generalization to input information with variable timescales.

## Efficient execution on neuromorphic hardware

In recent years, various neuromorphic platforms have been developed for SNNs, which help achieve higher execution efficiency than general-purpose platforms such as CPUs and GPUs. Compared to ordinary SNNs with only soma dynamics, DH-SNNs additionally involve the computation of dendritic dynamics, which make them difficult to operate on conventional neuromorphic hardware. We have developed several hybrid-paradigm neuromorphic chips during the past ten years, Tianjic series[39,40], which can support ANNs, SNNs, and hybrid neural networks[41], thus providing the possibility of performing DH-SNNs by configuring the spiking mode for soma dynamics and the non-spiking mode for dendritic dynamics. In this subsection, we deploy DH-SNNs on a recent Tianjic chip, TianjicX[40], to demonstrate the feasibility of efficient execution of DH-SNNs on domain-specific hardware. Fortunately, more and more neuromorphic chips such as Loihi 2(https://download.intel.com/newsroom/2021/new-technologies/neuromorphic-computing-loihi-2-brief.pdf), SpiNNaker 2[42] and BrainScale 2[43] have adopted hybrid-paradigm idea, which indicates DH-SNNs have great potential in applying to practical neuromorphic systems.

To better utilize the resources of TianjicX, we add an extra restriction on the synaptic connection pattern as illustrated in Fig. 6d.

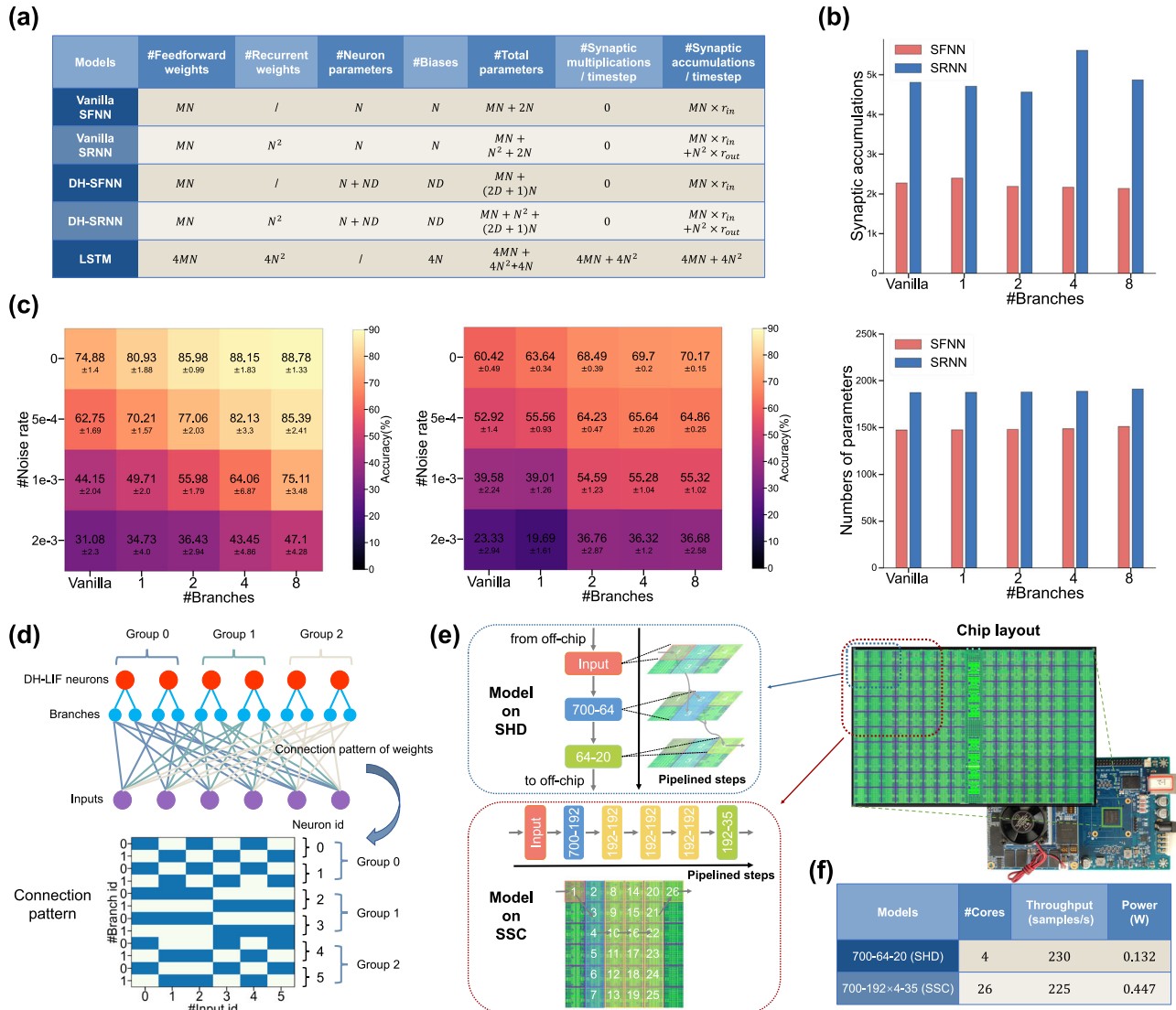

**Fig. 6 | Model compactness, robustness and efficient execution on neuromorphic hardware. a** Theoretical number of parameters and synaptic operations of vanilla SNNs, DH-SNNs, and LSTM. We assume that a layer has $N$ neurons with $M$ inputs. $r_{in}$ and $r_{out}$ represents the mean firing rates of spike inputs and outputs, respectively. Synaptic multiplications and accumulations only include the computation of weight matrices. **b** Comparing the number of synaptic accumulations and parameters of vanilla SNNs and DH-SNNs with different numbers of dendritic branches. **c** Comparing robustness of vanilla SFNNs and DH-SFNNs in resisting random spike noises on SHD (left) and SSC (right) datasets. **d** Illustration of the synaptic connection pattern of DH-SNNs for deployment on neuromorphic hardware, where neurons within each group share the same pattern for easier mapping without degrading much accuracy. **e** The TianjicX development board and the dataflow when performing DH-SNNs on SHD and SSC datasets. The model on SHD uses four functional cores with three timing phase groups and the model on SSC uses 26 functional cores with six timing phase groups. Multiple timing phase groups are scheduled in a pipelined manner. **f** The execution performance including throughput and dynamic power consumption when performing DH-SNNs on the TianjicX neuromorphic chip at 400 MHz clock frequency. Notice that processing one sample takes 1000 timesteps. The standard deviations (presented as error bars) represent 5 repeated trials.

Specifically, the continuous neurons in a layer within a neuron group share one synaptic connection pattern on the dendritic branches with the same branch index. For example, in Fig. 6d, branch 0 of neuron 0 and branch 0 of neuron 1 are connected to the same synaptic inputs. In this way, the synaptic operations of these two branches can be performed together. In our implementation, the neuron group size is set to 32, so here we modify the number of neurons in each hidden layer to be integer multiples of 32. Two DH-SNN models with the above connection pattern restriction are implemented on the TianjicX neuromorphic chip. One model is a single-layer DH-SFNN on the SHD dataset, and the other is a four-layer DH-SFNN on the SSC dataset. We have compared the two models with and without the connection pattern restriction and found the restriction only induces negligible accuracy degradation within 0.3%. The single-layer DH-SFNN only uses four of a chip's 160 functional cores, while the four-layer DH-SFNN uses 26 functional cores. We divide each model into several execution steps and allocate different numbers of functional cores to them as presented in Fig. 6e. The flexible timing schedule of TianjicX enables a pipelined execution of the steps for better performance. As summarized in Fig. 6f, both DH-SNNs can be efficiently performed on TianjicX with high throughput and low power consumption. More details of hardware implementation are provided in Methods and Supplementary Fig. S11.

## Application to EEG signal recognition and robot place recognition

In the field of brain-computer interface, how to handle electroencephalogram (EEG) signals effectively is a significant problem.

**Table 1 | Accuracy comparison between DH-SNNs and prior methods**

| Dataset | Model | #Parameters | Accuracy |
|---|---|---|---|
| SHD | SFNN[38] | 0.09M | 48.1% |
| | SRNN[38] | 1.79M | 83.2% |
| | SRNN[64] | 0.17M | 81.6% |
| | SRNN[31] | 0.11M | 82.7% |
| | SCNN[65] | 0.21M | 84.8% |
| | SRNN[33] | 0.14M | 90.4% |
| | LSTM[38] | 0.43M | 89.2% |
| | **DH-SRNN (1-layer, 2-branch)** | **0.05M** | **91.34%** |
| | **DH-SFNN (2-layer, 8-branch)** | **0.05M** | **92.1%** |
| SSC | SFNN[38] | 0.09M | 32.5% |
| | SRNN[31] | 0.11M | 60.1% |
| | SRNN[33] | 0.77M | 74.2% |
| | LSTM[38] | 0.43M | 73.1% |
| | **DH-SFNN (4-layer, 4-branch)** | **0.27M** | **81.03%** |
| | **DH-SRNN (3-layer, 4-branch)** | **0.35M** | **82.46%** |
| S-MNIST | LSNN[66] | 0.08M | 96.4% |
| | AHP-SNN[67] | 0.08M | 96.0% |
| | SRNN[33] | 0.16M | 98.7% |
| | LSTM[68] | 0.06M | 98.2% |
| | **DH-SRNN (2-layer, 2-branch)** | **0.08M** | **98.9%** |
| PS-MNIST | LSTM[68] | 0.06M | 88% |
| | SRNN* (not standard inputs)[33] | 0.16M | 94.3% |
| | **DH-SRNN (2-layer, 1-branch)** | **0.08M** | **94.52%** |
| GSC | SRNN[69] | 0.04M | 86.7% |
| | LSNN[70] | 4.19M | 91.2% |
| | SRNN[33] | 0.31M | 92.1% |
| | **DH-SRNN (1-layer, 8-branch)** | **0.13M** | **93.86%** |
| | **DH-SFNN (3-layer, 8-branch)** | **0.11M** | **94.05%** |
| TIMIT | LSNN[66] | 0.4M | 66.8% |
| | LSNN[71] | 0.4M | 65.4% |
| | SRNN[33] | 0.63M | 66.1% |
| | **DH-SRNN (1-layer, 8-branch)** | **0.18M** | **67.42%** |

*The bolded portion in the table represents the results of this study.

Existing approaches include conventional classification algorithms[44,45] and emerging deep learning-based algorithms such as convolutional neural networks (CNNs)[46] and RNNs[47]. Recently, SNN-based methods[48,49] also show great potential in processing EEG signals with high efficiency but have not achieved satisfactory performance yet. Considering the intrinsic multi-timescale components in EEG signals, we believe our proposed DN-SNNs can boost the performance of SNNs in EEG signal recognition tasks.

We select an EEG-based emotion recognition task with the DEAP dataset[50] to evaluate DH-SNNs. As illustrated in Fig. 7a, the DEAP dataset contains EEG signals recorded by electrodes from 32 participants stimulated with music videos. Along with EEG signals, participants were asked to report their emotions while watching music videos, using as the label for emotion recognition. After pre-processing (see Methods), EEG signals were fed to one-layer DH-SFNNs with different numbers of dendritic branches. We use DH-SFNNs to recognize three levels (low, medium and high) of valence and arousal which reflect emotion on the DEAP dataset (see Methods for more details). The accuracy curves during model training are provided in Fig. 7b, c. We find that DH-SFNNs show much better performance than vanilla SFNNs in both tasks. Consistently, the temporal dendritic heterogeneity under multiple dendritic branches indeed helps boost performance, which evidences our

prediction that DH-SNNs have great potential in processing multi-timescale EEG signals. As summarized in Supplementary Table S6, DH-SNNs once more demonstrate the best recognition accuracy on the DEAP dataset with much fewer parameters compared to existing approaches including multi-layered perceptron (MLP)[51], CNN[46], and spiking CNN (SCNN)[48]. In Supplementary Fig. S12, we additionally conduct similar experiments with SRNNs and two-class emotion recognition, where the above conclusions still hold. Compared to the results with DH-SFNNs, we observe higher accuracy with DH-SRNNs but a reduced accuracy gap when varying the number of dendritic branches. This evidences again the faster performance saturation of SRNNs.

We then design a visual place recognition (VPR) task to demonstrate the potential of our model in the field of robots. The robot visual place recognition has become an increasingly important area in the robotics community, as it enables robots to better comprehend the spatial properties of the environment[52]. Currently, there are two primary approaches being explored for visual place recognition. The former uses temporally captured images for place recognition, such as SeqSLAM[53], FlyNet[54] and sequential place learning[55], while the latter[56,57] uses neuromorphic sensors such as event cameras[7,8] as an extra data source for improving recognition accuracy. In our experiments, we design a NeuroVPR task and use a mobile robot to collect the spike event stream while moving in the indoor environment. The target is to recognize where it is using the collected spike event stream. The details of the dataset and the experiment setting can be found in Methods. We compare the performance of our DH-SNN model to the vanilla SNN model. The results in Supplementary Fig. S13 demonstrate higher top-1, top-5, and top-10 accuracy scores of the DH-SNN model, which shows great potential in performing robotic tasks with rich temporal information. Notice that here SRNNs do not show better performance than SFNNs, which might be due to the differences in network architectures and the training difficulties of recurrent networks in image recognition tasks.

## Discussion

We propose the DH-LIF neuron model which incorporates temporal dendritic heterogeneity into the spiking neuron and then extend to the network level for constructing DH-SFNNs and DH-SRNNs. By learning heterogeneous timing factors on different dendritic branches through the adapted BPTT algorithm, DH-SNNs are able to extract, memorize, and integrate temporal features at different timescales. This rich temporal heterogeneity significantly improves the comprehensive performance of SNNs in terms of accuracy, compactness, robustness, and generalization when performing temporal computing benchmarks we validated for speech recognition, visual recognition, EEG signal recognition, and robot place recognition. Owing to the additional sparse restriction on the connection pattern, DH-SNNs do not increase storage and computational costs, allowing efficient execution on neuromorphic hardware. This work demonstrates a potential route to exploit biological observations appropriately for moving neuromorphic computing a big step toward real-world applications.

The above metrics are easy options to select for measuring the performance of DH-SNNs, however, they are not intuitive for understanding the underlying working mechanism. To this end, we elaborate on a delayed spiking XOR problem as a simple but clear benchmark for the proposed DH-SNNs. In the naive delayed spiking XOR problem, we demonstrate the long-term memory of each dendritic branch without state reset like the membrane potential. In the multi-timescale spiking XOR problem, we reveal that different dendritic branches with variable timing factors can capture multi-timescale temporal features, for example simultaneously memorizing low-frequency signals and tracking high-frequency signals, enabling combined decisions at the soma through feature integration. Furthermore, we reveal that the network-level connections including inter-layer feedforward connections and intra-layer recurrent connections can also integrate features

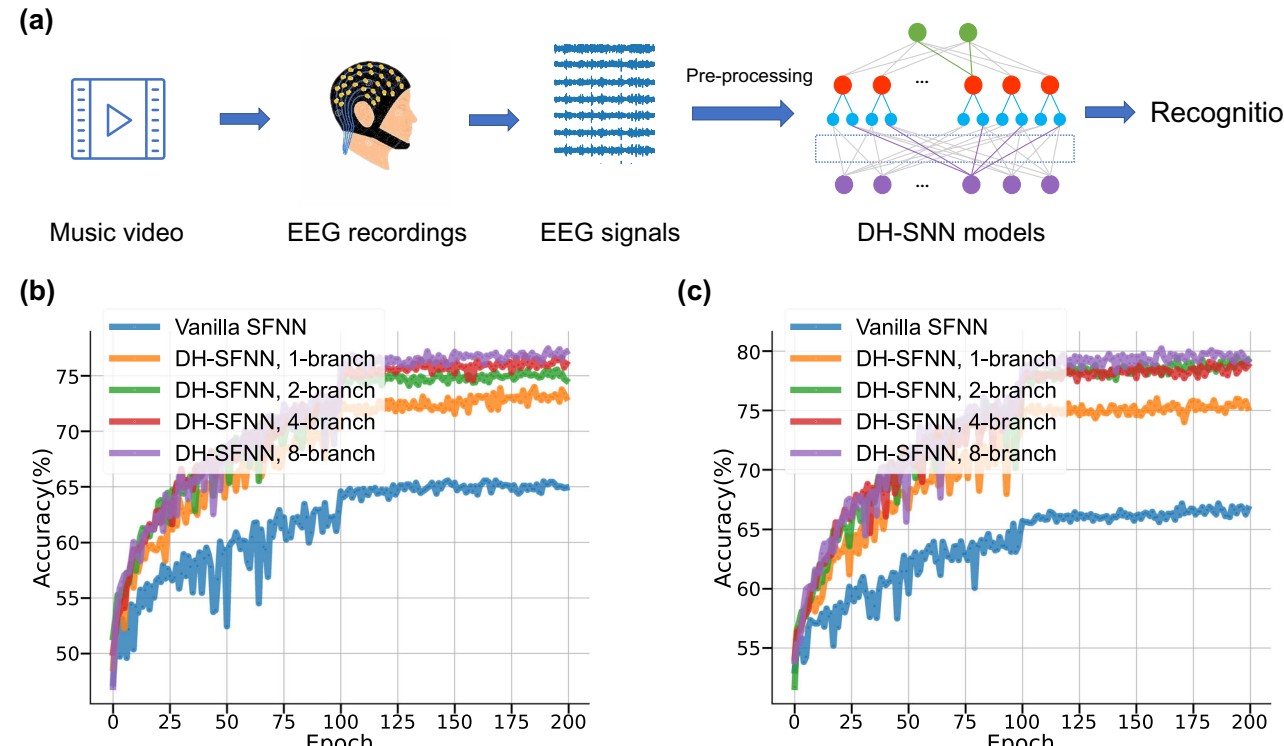

**Fig. 7 | Applying DH-SNNs to EEG signal recognition. a** The processing pipeline of DH-SNNs for EEG-based emotion recognition. Recognition accuracy curves of one-layer DH-SFNNs in three-class (**b**) valence and (**c**) arousal emotion recognition tasks on the DEAP dataset.

to produce high-level temporal features for making more complicated decisions. Usually, appropriately more dendritic branches generate richer dendritic temporal heterogeneity that enhances the representation power of DH-SNNs. Due to the higher complexity of feature integration given by recurrent connections, we observe faster performance saturation in DH-SRNNs compared to DH-SFNNs when performing the same task as the number of dendritic branches or layers grows. Comprehensively considering the above experimental results, we explain the working mechanism of temporal dendritic heterogeneity in DH-SNNs for performing multi-timescale temporal computing tasks: the inter-branch feature integration in a neuron, the inter-neuron feature integration in a recurrent layer, and the inter-layer feature integration in a network have similar and synergetic effects in capturing multi-timescale temporal features.

Overall, the proposed DH-SNN model is simple but quite effective as evidenced by extensive experiments. An interesting topic in future work is to improve the model itself. The current modeling is based on the LIF neuron model, which is the simplest form of spiking neurons even though it is widely used. A possible way for model improvement is to build DH-SNNs based on more complicated spiking neuron models rather than the LIF one. For example, neuron models with more dendritic properties found in biological neurons seem promising. However, the naive imitation of biological neurons may not benefit the performance of neural networks in practical tasks but even be harmful under the current intelligence framework due to the complicated equations with massive hyper-parameters needed for describing dendritic behaviors[32]. Therefore, an elaborate abstraction of dendritic properties like nonlinearity and careful transformation is the key to the success of neuron model exploration. Recent works[58,59] proposed an efficient spike-driven learning method based on dendritic computation serving as the adjustment to synaptic learning rules and further implemented them on FPGA, demonstrating a positive example in this regard. Another potential direction is to explore a learnable dendritic connection pattern. In contrast to the fixed dendritic connection

pattern in our modeling, biological neural networks exhibit evolving connections on dendrites. Drawing inspiration from this biological phenomenon, we can investigate the potential for adapting the connection pattern during the learning process. For instance, we can leverage methods like DEEP R[60] to automatically modify the network's connection pattern by pruning and rewiring synapses according to their significance. Moreover, well-designed optimization methods and appropriate benchmarking tasks are also critical for mining the potential of neuron models, which are left for future exploration. In addition, because we focus on demonstrating the effectiveness of temporal dendritic heterogeneity and revealing its working mechanism in this work, we select the simple fully-connected rather than convolutional layers as the backbone and do not pay much attention to training optimization techniques. This is the reason that we exclude the comparison with prior CNNs in most testing cases. It is quite possible to further improve the model performance if we introduce the convolutional topology and some optimization techniques such as activity normalization[3].

There are many inherent constraints in biological neurons. However, our research primarily centers on effectively integrating biological observations into computational models to solve real-world computing tasks, rather than strictly adhering to all biological principles. In fact, many works on bio-inspired algorithms did not follow strict biological constraints. For example, they extend the range of the timing factors of membrane potentials[31,61], and represent individual neurons using abstract units that communicate through continuous firing rates instead of discrete action potentials[62]. These departures from biological fidelity are often necessary to prevent the degradation of model performance during training with complicated neural dynamics and biological details. In our work, the timing factors are not unbounded and we restrict them within $(0, 1)$ through the *sigmoid*($\cdot$) function (see Equation (6) in Methods), but this is not the result of considering biological constraints. It is very hard to balance the performance in practical tasks and the biological plausibility. Innovations

in learning algorithms offer promising potential to realize this balance, which is an interesting topic for future work.

Processing temporally heterogeneous information is an important capability of not only the brain but also man-made machines. For example, a robot embedded with multimodal sensors must sense and process input signals with rich spectral components to make prompt and correct decisions. Besides improving the proposed model as aforementioned, applying the model to real-world complex scenarios and deploying it on practical agents are promising future work, which will bridge neuroscience and reality more clearly. At that time, visualization and analysis of the interactions between different modalities from the perspective of neural dynamics would be of interest and helpful for understanding how the brain processes multimodal information concurrently and efficiently.

# Methods

## Modeling dendritic memory

The dendrite structure of a spiking neuron can be regarded as a series of small RC circuits where the current $i_d^{t,x}$ and voltage $u_d^{t,x}$ on a dendritic branch vary over time and location following complex differential equations[63], which is usually neglected in popular LIF neuron models. In order to ease the implementation on computers, we simplify the model for friendly programming. Specifically, we consider each dendritic branch as a whole RC circuit while removing the spatial dendritic features and only keeping the temporal features. The dendritic current $i_d^t$ can behave as

$$I_{d\_ext}^t = C\frac{\partial u_d^t}{\partial t} + \frac{u_d^t}{R_T} + i_d^t, \quad u_d^t = R_L i_d^t \tag{1}$$

where $I_{d\_ext}^t$ is the external synaptic current injected into the dendritic branch, $C$ and $R_T$ represent the equivalent capacitance and resistance, respectively, and $R_L$ denotes the connecting resistance between the dendrite and the soma. Then, we have

$$\tau_d\frac{\partial i_d^t}{\partial t} = -i_d^t + I_d^t \tag{2}$$

where $\tau_d = CR_L R_T/(R_L + R_T)$ represents the time constant of the dendritic branch and $I_d^t = I_{d\_ext}^t R_T/(R_L + R_T)$ denotes the synaptic inputs. If we discretize the above equation using the Euler method, we can have two formats

$$\tau_d(i_d^t - i_d^{t-1}) = -i_d^t + I_d^t \text{ or } \tau_d(i_d^{t+1} - i_d^t) = -i_d^t + I_d^t. \tag{3}$$

Thus, the discrete versions can be written as

$$i_d^{t+1} = \alpha_d i_d^t + (1 - \alpha_d)I_d^{t+1} \tag{4}$$

where the timing factor $\alpha_d$ equals $1 - \frac{1}{\tau_d + 1}$ or $\alpha_d = 1 - \frac{1}{\tau_d}$ according to the two discretization formats, respectively. Although there are two different definitions of $\alpha_d$ with respect to $\tau_d$, they share a unified representation if we only look at the $\alpha_d$ level, which is just the reason that we learn $\alpha_d$ rather than $\tau_d$ in our experiments.

## LIF-based spiking neuron with dendritic heterogeneity (DH-LIF)

With the modeling of the memory on each dendritic branch, we redesign the classic LIF-based spiking neuron model. The classic LIF neuron only has single-timescale memory on the soma's membrane potential, while the DH-LIF neuron further has multi-timescale memories on the dendrite. The behaviors of a DH-LIF neuron can be governed by

$$\begin{cases} u^{t+1} = \beta u^t + (1 - \beta)\sum_d R i_d^{t+1} - o^t u_{th} \\ i_d^{t+1} = \alpha_d i_d^t + (1 - \alpha_d)I_d^{t+1} \\ o^{t+1} = H(u^{t+1} - u_{th}) \end{cases} \tag{5}$$

where $u$ is the soma's membrane potential, $\beta$ is the timing factor of the membrane potential, $R$ is the soma's membrane resistance which is set to $R = 1$ for simplification, $d$ is the index of dendritic branches, and $u_{th}$ is the firing threshold of the membrane potential. $H(\cdot)$ is the Heaviside function that follows $H(x) = 1$ when $x \geq 0$ and $H(x) = 0$ otherwise. When the neuron fires a spike, the membrane potential decreases by $u_{th}$. To avoid negative timing factors in Equation (5), $\alpha_d$ and $\beta$ should be restricted within $[0, 1]$, which is realized by adding a sigmoid function for soft clamping:

$$\alpha_d = sigmoid(\hat{\alpha}_d), \quad \beta = sigmoid(\hat{\beta}). \tag{6}$$

The synaptic input on the $d$-th dendritic branch is the sum of the feedforward input and the recurrent input:

$$I_d^{t+1} = \langle \mathbf{W}_d, \mathbf{X}^{t+1}\rangle + \langle \mathbf{U}_d, \mathbf{o}^t\rangle \tag{7}$$

where $\mathbf{W}_d$ and $\mathbf{U}_d$ represent feedforward and recurrent synapse vectors, respectively, which are two sparse vectors since only the synapses connected to the $d$-th dendritic branch are valid.

## SNN with DH-LIF neurons (DH-SNN)

Extending the DH-LIF neuron model to an SNN model (DH-SNN), we first add the layer information into Equation (5) and get the dynamics of an SNN layer as follows

$$\begin{cases} \mathbf{u}^{t+1,l} = \boldsymbol{\beta}^l \odot \mathbf{u}^{t,l} + (1 - \boldsymbol{\beta}^l) \odot R\sum_d \mathbf{i}_d^{t+1,l} - \mathbf{o}^{t,l}u_{th} \\ \mathbf{i}_d^{t+1,l} = \boldsymbol{\alpha}_d^l \odot \mathbf{i}_d^{t,l} + (1 - \boldsymbol{\alpha}_d^l) \odot \mathbf{I}_d^{t+1,l} \\ \mathbf{o}^{t+1,l} = H(\mathbf{u}^{t+1,l} - u_{th}) \end{cases} \tag{8}$$

where $l$ denotes the layer index and $\odot$ represents the element-wise multiplication. Then, the synaptic current on the $d$-th dendritic branch can be

$$\mathbf{I}_d^{t+1,l} = \mathbf{W}_d^l \mathbf{o}^{t+1,l-1} + \mathbf{U}_d^l \mathbf{o}^{t,l} \tag{9}$$

where $\mathbf{W}_d$ and $\mathbf{U}_d$ denote the matrix forms of feedforward and recurrent synaptic weights, respectively, which are again sparse since only the synapses connected to the $d$-th dendritic branches of neurons in the $l + 1$ layer are valid.

## Sparse connection restriction

Usually, the topology at the network level has two cases: with or without recurrent connections. We term the DH-SNN with only feedforward connections as DH-SFNN and the one with recurrent connections as DH-SRNN. We assume that a layer with $N$ neurons has $M$ inputs, then we have $\mathbf{W}_d \in N \times M$ and $\mathbf{U}_d \in N \times N$. From the perspective of the $n$-th neuron, its synaptic weight matrix connected to feedforward and recurrent inputs can be denoted as $\mathbf{W}_n \in D \times M$ and $\mathbf{U}_n \in D \times N$, respectively. Briefly, $\mathbf{W}_n$ and $\mathbf{U}_n$ are respectively assembled by the $n$-th row of $\mathbf{W}_d$ and $\mathbf{U}_d$, $d = 0, 1, 2, ..., D-1$.

In a layer of the DH-SFNN, there are only feedforward connections, i.e., Equation (9) becomes $\mathbf{I}_d^{t+1,l} = \mathbf{W}_d^l \mathbf{o}^{t+1,l-1}$. we restrict the

connections of each neuron as follows

$$\sum_d |P_d| = M, \quad \cup_d P_d = \{0,1,\ldots,M-1\}, \quad |P_d| \approx \left\lceil \frac{M}{D} \right\rceil \tag{10}$$

where $P_d$ denotes the index set of feedforward synapses connected to the $d$-th dendritic branch, i.e., the set of non-zero elements in the $d$-th row in $\mathbf{W}_n$. $|P_d|$ denotes the set size. Similarly, for a layer of the DH-SRNN, we restrict both feedforward and recurrent connections as follows

$$\begin{cases} \sum_d |P_d| = M, \quad \cup_d P_d = \{0,1,\ldots,M-1\}, \quad |P_d| \approx \left\lceil \frac{M}{D} \right\rceil \\ \sum_d |Q_d| = N, \quad \cup_d Q_d = \{0,1,\ldots,N-1\}, \quad |Q_d| \approx \left\lceil \frac{N}{D} \right\rceil \end{cases} \tag{11}$$

where $Q_d$ applies similar definitions with $P_d$ on recurrent synapses. From the above equations, it can be seen that our solution would not increase connection and computational costs as the number of dendritic branches grows. The number of synapses on each dendritic branch is balanced to a great extent by limiting the connections to a sparse pattern. In this way, the DH-SNN maintains the lightweight computational advantage of ordinary SNNs.

## Learning of DH-SNN

In order to achieve high performance, we adopt the emerging SNN-version BPTT learning algorithm[33] and extend it from ordinary SNNs to our DH-SNNs. The model parameters including synaptic weights, $\mathbf{W}, \mathbf{U}$, and timing factors, $\hat{\boldsymbol{\alpha}}, \hat{\boldsymbol{\beta}}$, are automatically learned during training. Assuming the loss function is $L$ and applying the chain rule of the gradient descent, the BPTT for the DH-SNN can follow

$$\begin{cases} \delta \mathbf{u}^{t,l} = \boldsymbol{\beta}^l \odot \delta \mathbf{u}^{t+1,l} + H' \odot \delta \mathbf{o}^{t,l} \\ \delta \mathbf{i}_d^{t,1} = (1-\boldsymbol{\beta}^l) R \odot \delta \mathbf{u}^{t,l} + \boldsymbol{\alpha}_d^l \odot \delta \mathbf{i}_d^{t+1,1} \\ \delta \mathbf{o}^{t,l} = -u_{th} \delta \mathbf{u}^{t+1,l} + \sum_d \mathbf{W}_d^{l+1^T}(1-\boldsymbol{\alpha}_d^{l+1}) \odot \delta \mathbf{i}_d^{t,l+1} \\ \quad + \sum_d \mathbf{U}_d^{l^T}(1-\boldsymbol{\alpha}_d^l) \odot \delta \mathbf{i}_d^{t+1,l} \end{cases} \tag{12}$$

where $\delta$ denotes the gradient of the loss function $L$ with respect to specific variables. Note that $H'$ actually does not exist due to the non-differentiable spiking activities. To address this issue, we adopt the widely used surrogate gradient but replace the hard rectangle approximate curve with a soft multi-Gaussion curve[33]:

$$H' = \frac{\partial o^t}{\partial u^t} = \gamma(1+h)\mathcal{N}(u^t|u_{th},\sigma^2) - \gamma h\mathcal{N}(u^t|\sigma,(s\sigma)^2) - \gamma h\mathcal{N}(u^t|-\sigma,(s\sigma)^2) \tag{13}$$

where $\gamma$, $h$ affect the magnitude and $\sigma$, $s$ affect the width of the gradient. The peak of the surrogate gradient function is at the firing threshold $u_{th}$ where the neuron fires a spike. Finally, the gradients of parameters can be achieved by

$$\begin{cases} \delta \mathbf{W}_d^l = \sum_t (1-\boldsymbol{\alpha}_d^l) \odot \delta \mathbf{i}_d^{t,l} \mathbf{o}^{t,l-1^T}, \quad \delta \mathbf{U}_d^l = \sum_t (1-\boldsymbol{\alpha}_d^l) \odot \delta \mathbf{i}_d^{t+1,l} \mathbf{o}^{t,l^T} \\ \delta \hat{\boldsymbol{\beta}}^l = \sum_t \delta \hat{\boldsymbol{\beta}}^{t,l} = \sum_t \delta \boldsymbol{\beta}^{t,l} \odot (1-\delta \boldsymbol{\beta}^{t,l}), \quad \delta \boldsymbol{\beta}^{t,l} = \mathbf{u}^{t-1,l} \odot \delta \mathbf{u}^{t,l} - R \sum_d \mathbf{i}_d^{t,l} \odot \delta \mathbf{u}^{t,l} \\ \delta \hat{\boldsymbol{\alpha}}_d^l = \sum_t \delta \hat{\boldsymbol{\alpha}}_d^{t,l} = \sum_t \delta \boldsymbol{\alpha}_d^{t,l} \odot (1-\delta \boldsymbol{\alpha}_d^{t,l}), \quad \delta \boldsymbol{\alpha}_d^{t,l} = \mathbf{i}_d^{t-1,l} \odot \delta \mathbf{i}_d^{t,l} - \mathbf{I}_d^{t,l} \odot \delta \mathbf{i}_d^{t,l} \end{cases} \tag{14}$$

## Datasets and tasks

The self-designed spiking XOR problem has two types of input spike patterns with high or low firing rates. We set the high-firing-rate pattern with a firing probability of 0.6 and the low-firing-rate pattern with

a firing probability of 0.2. Each spike pattern lasts $10ms$ and the length of each timestep in the simulation is $1ms$ in both the delayed spiking XOR problem and the multi-timescale spiking XOR problem. Specifically, in the multi-timescale spiking XOR problem, we set the time interval between two input spike patterns to $5ms$ for Signal 2 with faster periods. In addition, we added spike noises with a firing probability of 0.01 in the duration of experiments. For the spiking XOR problems, we run the experiments with 10 repeated trials.

Besides the self-designed spiking XOR problems, we also test our models on standard benchmarks. Spiking Heidelberg digits (SHD) and spiking speech command (SSC) datasets convert the original audio data into the spike format through a bionic inner ear model. SHD contains about 10,000 high-quality recordings of English and German speech for digits ranging from 0 to 9. A total of 12 speakers are included in the dataset, in which 6 are female and 6 are male. The speakers range in age from 21 to 56, with an average of 29 years old. Each speaker records about 40 sequences for each language and each digit, producing a total of 10,420 sequences. Each recording is clipped by a threshold associated with each speaker, which is optimized by a black-box optimizer. Further processing applies a fast Fourier transform and a $30ms$ Hanning window. The SSC dataset is derived from the Google speech command dataset (GSC 0.02 version). Each sample consists of a $1s$ audio file of a spoken English word with a sampling rate of 16 KHz. The whole dataset contains 105,829 audio files with 35 classes. Likewise, a $30ms$ Hanning window is applied at the beginning and the end of each audio recording before the spike conversion. We further pre-processed the raw spike data before feeding it into downstream networks. Specifically, we sampled the original spike trains with the time interval of $dt$, and truncated the original data according to the maximum time $T_{max}$. Each recording is converted into a $700 \times T$ matrix, where $T = T_{max}/dt$ is the number of total timesteps. The $i$-th column of the matrix is a vector with a length of 700, recording whether the channel emits spikes during $[(i-1)dt, idt)$. If there is a spike or more in the duration, the corresponding value of the channel is 1, otherwise is 0. The above datasets are divided into several sets such as training, testing, and validation sets. In particular, the SHD training and testing sets contain 8,156 and 2,264 pieces of data, respectively; the SSC training, testing, and validation sets contain 75,466, 9,981, and 20,382 pieces of data. For the SHD and SSC datasets, we run the experiments with 5 repeated trials.

S-MNIST and PS-MNIST datasets are based on the handwritten digit dataset, MNIST, for image recognition tasks. In S-MNIST, each $28 \times 28$ image in the original MNIST dataset is converted into a pixel sequence of length 784. Each time a pixel comes to the model, the neural network needs to memorize a time series of length 784 and then finally classify the input handwritten digit. In PS-MNIST, all pixel sequences are shuffled before being injected into the model, which increases the memorization and classification difficulty compared to S-MNIST. In essence, S-MNIST and PS-MNIST datasets are two important standard benchmarks for sequence learning and are mainly used to evaluate the long-term memory capability of spatiotemporal networks. For these datasets, the real-value inputs are directly fed into DH-SNNs. The first layer of DH-SNNs receives real-value inputs rather than spiking inputs, but still performs the spiking neural dynamics as a normal DH-SNN layer except for the different input format. In this way, the first layer actually acts as an encoding layer that converts non-spiking inputs to spiking outputs and then sends to post-synaptic layers. This encoding scheme also works for all following non-spiking datasets.

The GSC dataset v.1 contains 64,727 utterances from 1881 speakers saying 35 different speech commands. In our experiments, we followed the dataset setting as other SNN methods that transform the 30 classes of the dataset into 12 classes including ten words: "Yes", "No", "Up", "Down", "Left", "Right", "On", "Off", "Stop", "Go", and an additional special class named "Unknown" covering the left 25 classes

with an extra class "Silence" extracted randomly from the background noise audio files. We used an existing feature extraction method[33], i.e., adopting log Mel filters and extracting their first three derivative orders from the raw audio files by calculating the logarithm of 40 Mel filters coefficients using the Mel scale between 20 Hz and 4 KHz for pre-processing. Each frame of the inputs has $40 \times 3$ channels. The spectrograms are normalized and the length of each timestep in the simulation is $10ms$. Thus, each audio sample is transformed into a sequence of 101 frames with 120 channels. The TIMIT dataset contains acoustic speech signals of sentences spoken by 630 speakers from 8 major dialect regions of the United States. The goal of the tasks is to recognize the phonemes of every $10ms$ frame in each sentence. We follow the experimental settings of a prior work[33]. The training, validation, and testing sets contain 3,696, 400, and 192 sequences, respectively. The performance is evaluated on the core testing set. The raw audio data are pre-processed into 13 Mel Frequency Cepstral Coefficients (MFCCs) and then converted into 39 input channels including the first- and second-order derivatives and their combinations. Each frame belongs to 61 classes of phonemes. We set the simulation time interval of DH-SNNs to $1ms$, so every frame of a $10ms$ input signal is fed into the model for 10 consecutive timesteps.

The DEAP dataset contains 32-channel EEG data and 8-channel peripheral physiological signals recorded by the electrode arrays from 32 participants when watching 40 pieces of one-minute music videos. Before watching the music video, each participant experienced a $3s$ period of baseline recording time during which a fixation cross is presented on the screen. The EEG signals were sampled at 512 Hz and then downsampled to 128 Hz while removing electrooculography (EOG) artifacts for pre-processing. In our experiments, only the 32-channel EEG data was used for classification. Therefore, the dimension of total EEG data is $32 \times 40 \times 32 \times 8064$ (#participants × #trials × #channels × temporal length). After watching the music video, each participant was asked to report their emotion in levels of arousal, valence, liking, and dominance from 1 to 9. Among them, arousal ranges from inactive (e.g., uninterested, bored) to active (e.g., alert, excited), while valence ranges from unpleasant (e.g., sad, stressed) to pleasant (e.g., happy, elated). In our experiments, we use two emotion description dimensions, arousal and valence, and then map the score of 1 to 9 to three labels: low (score lower than 4), medium (score between 4 and 6), and high (score higher than 6), or to two labels: low (score lower than 5) and high (score higher than 5). In this way, each trial has its label of valence and arousal for recognition. The total EEG data were pre-possessed again before feeding to models. We follow the pre-possessing method used by Tao et al.[47]. The first $3s$ data of each trial is used for producing the average $1s$ baseline signal by averaging the $3s$ data per second. The following $60s$ data was normalized by subtracting the average $1s$ baseline signal every second and then divided into 20 segments of $3s$ for each. Finally, the dimension of total EEG data is transferred to $25600 \times 32 \times 384$ (#samples × #channels × temporal length). We choose 90% of the data as the training set and the remaining 10% as the testing set.

The NeuroVPR dataset for the robot visual place recognition task was collected by a Clearpath Jackal robot in the laboratory environment. A DAVIS 346 event camera is deployed on the robot platform, collecting frame-based images and spike event streams in real-time while the robot moves a $300m$ trajectory in the laboratory room. The visual resolution is $240 \times 346$. The dataset consists of a total 10 repeated trajectories recorded at night. In each trajectory, the robot starts and stops at the same position and follows the same route. The dataset consists of a total $1475 \times 10$ RGB frames and about 1.7 billion spike events. The goal of the task is to make the robot recognize its position through a period of visual signals on the running track. Notice that we only use the spike events in this task. In our experiments, we divide each trajectory into 100 segments with uniform length and mark the corresponding events in each segment with the same label. The spike events in the duration of $21ms$ were fed into the SNN models and generated the prediction which indicates the robot's position. We randomly chose six trajectories as the training set, three trajectories as the validation set, and one trajectory as the test set.

## Experimental setting

For all experiments, networks have two parts including stacked SNN layers and a following readout layer. For the self-designed spiking XOR problems and the NeuroVPR dataset, the readout layer is a simple linear layer that decodes the spike output of the last SNN layer into the possibility $\hat{y}_i^t$ of the $i$-th class at the $t$-th timestep. On SHD, SSC, GSC, TIMIT, and DEAP datasets, the readout layer is a non-spiking SNN layer with leaky membrane potentials and generates the possibility $\hat{y}_i$ of the $i$-th class by decoding the membrane potentials as $\hat{y}_i = \sum_t softmax(u_{i,out}[t])$ on SHD and GSC, and $\hat{y}_i = softmax(\sum_t u_{i,out}[t])$ on SSC, TIMIT, and DEAP. In the above tasks, we used a multi-Gaussian curve[33] following $\gamma, h, \sigma, s = 0.5, 0.15, 0.5, 6$ to approximate the gradient of the non-differentiable spike activity. On S-MNIST and PS-MNIST datasets, the readout layer is a vanilla SNN layer that decodes the spike output by counting spikes of each class, i.e., $\hat{y}_i = softmax(\sum_t o_{i,out}[t])$. On these two datasets, the hyper-parameters of the multi-Gaussian approximation curve are $\gamma, h, \sigma, s = 0.5, 0, 0.5, 0$. For the non-spiking datasets such as GSC, S-MNIST, and PS-MNIST, note that the input of the first SNN layer is non-spiking values. For all the above tasks, the loss function adopted is the Cross-Entropy loss following $L = -\sum_i y_i log\hat{y}_i$, where $\hat{y}_i$ represents the predicted possibility of the $i$-th class and $y_i$ is the ground truth. We used the Adam optimizer with an initial learning rate $10^{-2}$ and a step learning rate scheduler. The code is built with the Pytorch framework and executed on 8 NVIDIA RTX 3090 GPUs.

For DH-SNNs, there are many hyper-parameters to initialize, such as firing thresholds, membrane potential timing factors, and dendritic timing factors. For the detailed settings of hyper-parameters, network structures, and batch sizes please refer to Supplementary Information. Specifically, Supplementary Table S1 shows the model configuration details for the tasks used in ablation studies. In these tasks, since the network structure varies a lot, e.g., a single layer or multiple layers in SFNNs or SRNNs, we only offer the number of neurons per layer in the table while presenting details in Supplementary Fig. S1. Supplementary Table S2 shows the model configuration details for other tasks on standard temporal computing datasets. In these tasks, we do not implement comprehensive ablation but only show the best results with the network structures that can balance performance and efficiency. Therefore, we directly provide specific network structures in the table. Note that we use the DH-SRNN with a bidirectional structure, which consists of two parallel layers and receives inputs from both forward and backward directions. The spiking outputs of the two layers are then concatenated and fed to the decoding layer. Supplementary Table S3 presents the initialization configurations of the timing factors we used. Notice that the timing factors of each dendritic branch and the membrane potential, $\alpha_d$ and $\beta$, actually equal $sigmoid(\hat{\alpha}_d)$ and $sigmoid(\hat{\beta})$, respectively, wherein $\hat{\alpha}$ and $\hat{\beta}$ are the truly optimized parameters during training.

## Influence of the membrane potential reset mechanism

We analyze the influence of the membrane potential reset mechanism on the capability of long-term memory using several tasks. The tested neuron models include the vanilla LIF neuron with different reset mechanisms and the DH-LIF neuron. Three membrane potential reset mechanisms are selected: hard reset, soft reset, and without reset. The vanilla LIF neuron with the hard reset mechanism is widely used, which

is governed by

$$u^{t+1} = \beta(1 - o^t)u^t + (1 - \beta)i^{t+1}. \qquad (15)$$

The one with the soft reset mechanism can be described as

$$u^{t+1} = \beta u^t - o^t u_{th} + (1 - \beta)i^{t+1}. \qquad (16)$$

Last, the one without the reset mechanism follows

$$u^{t+1} = \beta u^t + (1 - \beta)i^{t+1}. \qquad (17)$$

In above equations, $u$ is the membrane potential, $\beta$ is the timing factor of the membrane potential, $i$ is the synaptic input, $o$ is the spike output, and $u_{th}$ is the firing threshold.

First, we test the long-term memorization capability of SFNNs based on the above LIF neuron models using the delayed spiking XOR problem. Results of vanilla SFNNs with different membrane potential reset mechanisms and DH-SFNNs are compared. As depicted in Supplementary Fig. S3a, the vanilla SFNNs with reset mechanisms perform the worst in this task. This is because reset mechanisms would periodically clear some temporal information stored in the soma's membrane potential. Especially, the hard reset mechanism would clear the information completely, making it fail to memorize information for the long term. The soft reset mechanism slightly alleviates this problem. Apparently, the removal of reset mechanisms or using the proposed DH-LIF neuron with dendritic memory can avoid this problem, thus significantly improving the long-term memorization capability. Next, we further test above models on SHD and SSC datasets under different sampling time intervals from $dt=8ms$ to $dt=1ms$. As presented in Supplementary Fig. S3b, c, vanilla SFNNs without the reset mechanism and DH-SFNNs still show better long-term memorization capability especially when processing slow-timescale data with $dt=1ms$. However, as the sampling time interval increases, the performance of vanilla SFNNs without the reset mechanism degrades quickly, which implies that it cannot handle fast-timescale information without any reset mechanism to clear the historic memory. Among these models, DH-SFNNs can generalize the best when processing information from the fast timescale to the slow timescale. This is owing to both the long-term memory of dendritic branches and the periodical clearing of historic information on the membrane potential. When the dendritic timing factors are small, the DH-SFNN would behave like the vanilla SFNN with the soft reset mechanism, performing well at the fast timescale, e.g., $dt=8ms$ here. By contrast, When the dendritic timing factors are large, the dendritic currents decay slowly with a long-term memory behaving like the membrane potentials of vanilla SFNNs without the reset mechanism, performing well at the slow timescale, e.g., $dt=1ms$ here.

### Influence of the dendritic connection pattern

To maintain the volume of parameters to a low level, we add a sparse restriction on the dendritic connections of DH-SNNs. In detail, we make each dendritic branch in a DH-LIF neuron only connect to a part of synaptic inputs. In this experiment, we vary the connection sparsity ratio ($s$) in a one-layer DH-SFNN with eight dendritic branches in each neuron and evaluate its performance on SHD and SSC datasets. Here the sparsity ratio represents the ratio of synaptic inputs connected to each dendritic branch over total synaptic inputs. Given an $s$ setting, we have

$$\sum_d |P_d| = M \cdot D \cdot s, \quad |P_d| \approx [M \cdot s] \qquad (18)$$

where $P_d$ denotes the index set of synaptic inputs connected to the $d$-th dendritic branch and $|P_d|$ denotes the set size. Here $P_d$ is determined by

$$P_d = L[k_d : (k_d + |P_d|)], \quad k_d = \left[\frac{M}{D}\right] \times d \qquad (19)$$

where $D$ represents the number of dendritic branches in each neuron and $L$ is a random sequence from 1 to $M$ which is different for each neuron. Therefore, when we set $s \le \frac{1}{D}$, there is no overlapped synaptic input between dendritic branches in each neuron. While when we set $s > \frac{1}{D}$, some synaptic inputs can connect to multiple dendritic branches in the meantime. In this case, the last $P_d$ will circularly read indexes from scratch for avoiding overflow.

The experimental results are shown in Supplementary Fig. S10. When the sparsity ratio is close to $\frac{1}{D}$ ($\frac{1}{8}$ here), the models perform well. On the contrary, when the sparsity ratio is too small or too large, the performance degrades to accuracy scores lower than 88% on SHD and lower than 69% on SSC. Generally speaking, on one hand, the synapse inputs connected to each neuron should better cover all synaptic inputs to get information as much as possible. On the other hand, the excessive overlap of synapse inputs connected to different dendritic branches would cause overfitting and performance degradation.

### Details of implementation on neuromorphic hardware

The TianjicX neuromorphic chip supports a hybrid-paradigm primitive instruction set that covers a wide range of operations. To implement DH-SNNs on the chip, only five operations listed in Supplementary Fig. S11a are necessary. Unlike ordinary SNNs, DH-SNNs require two types of LIF operations, one outputting binary spikes for soma dynamics and one outputting continuous currents for dendritic dynamics. In TianjicX, the spikes are represented by 2-bit ternary numbers while the membrane potentials are represented by 8-bit integers. The detailed configuration of LIF operation parameters can be found in Supplementary Table S2 of a recent reference[40].

The logical mapping of DH-SNNs onto functional cores in TianjicX is illustrated in Supplementary Fig. S11b. Each dashed box represents a functional core with operations and the dataflow. Taking the single-layer DH-SFNN on the SHD dataset using four functional cores (left) as an example, the functional core in the first timing phase group (Group 1), receives spikes from the host and multicasts them to the two functional cores in Group 2. Each functional core in Group 2 first divides the inputs into four parts corresponding to four dendritic branches, then every time performs an FC operation and a non-spiking LIF operation to simultaneously generate the dendritic currents with the same branch index of 32 neurons in a neuron group and finally executes a sum operation followed by a spiking LIF operation to generate the output spikes of the final 32 neurons. The last timing phase group contains only one functional core, which performs an FC operation and a non-spiking LIF operation based on the spikes collected from the previous group and sends the outputs to the host. For the four-layer DH-SFNN on the SSC dataset using 26 functional cores (right), the model mapping is quite similar to that of the single-layer DH-SFNN except for the different numbers of layers and neurons.

The operations in each functional core are executed in series, while the functional cores in the same timing phase group work in parallel. For inter-group execution, the timing phase groups are scheduled in a pipelined manner as illustrated in Supplementary Fig. S11c. In this way, the throughput can be improved, which depends on the timing phase group with the longest latency. For both models, the first timing phase group, i.e., Group 2, which processes the largest number of spike inputs, consumes the longest latency and determines the overall throughput. Notice that the specific clock cycle numbers in Supplementary Fig. S11c are acquired from the cycle-accurate chip simulator instead of real chip testing, so there exist certain errors compared to the physically measured results in Fig. 6f.

## Reporting summary

Further information on research design is available in the Nature Portfolio Reporting Summary linked to this article.

## Data availability

All data used in this paper are publicly available and can be accessed at http://yann.lecun.com/exdb/mnist/ for S-MNIST and PS-MNIST datasets, https://zenkelab.org/resources/spiking-heidelberg-datasets-shd/ for SHD and SSC datasets, https://tensorflow.google.cn/datasets/catalog/speech_commands/ for the GSC dataset. The TIMIT dataset is available on request via https://doi.org/10.35111/17gk-bn40. The DEAP dataset is available on request via https://www.eecs.qmul.ac.uk/mmv/datasets/deap/. The NeuroVPR dataset is available on Zenodo: https://doi.org/10.5281/zenodo.7825811.

## Code availability

The source code is publicly available at https://github.com/eva1801/DH-SNN.

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

## Acknowledgements

This work was partially supported by STI 2030 – Major Projects 2021ZD0200300, National Natural Science Foundation of China (No. 62276151, 62106119, 62236009, U22A20103), National Science Foundation for Distinguished Young Scholars (No. 62325603), CETC Haikang Group-Brain Inspired Computing Joint Research Center, and Chinese Institute for Brain Research, Beijing. We would like to thank Prof. Luping Shi for the valuable discussion.

## Author contributions

H.Z. and L.D. conceived the work. H.Z., R.H., and F.Y. carried out the simulation experiments. Z.Z. and X.L. carried out the hardware implementation. H.Z., Z.Z., and L.D. contributed to the analyses of experimental results. All of the authors contributed to the discussion of model and experiment design, and L.D. led the discussion. H.Z., Z.Z., B.X., Y.W., G.L., and L.D. contributed to the writing of the paper. L.D. supervised the whole project.

## Competing interests

The authors declare no competing interests.
