## [Peer Review File · Nature Communications]

REVIEWER COMMENTS

Reviewer #1 (Remarks to the Author):

This is a very interesting study by Zheng et al., where they investigate how heterogeneity, introduced by incorporating dendrites with different temporal scales, can be exploited for the processing of temporally variable signals. This work builds on the previous work of Perez-Nieves, et al. Nat. Communications 2021, and shows that spiking neural network (SNNs) models with temporal dendritic heterogeneity enable multi-timescale dynamics by learning heterogeneous timing factors on different dendritic branches. Specifically, the authors report higher accuracy, robustness, and generalization compared to vanilla SNNs. Importantly, they show that a fusion of features (similar to the multiplexing of Naud, Sprekeler PNAS 2018) is needed for performing multi-timescale tasks and that this fusion can be achieved either within a single neuron, between neurons in a recurrent layer, or between layers. This necessity further indicates that high performance in multi-timescale tasks cannot be simply achieved by increasing the number of parameters within a layer. The authors further seek to demonstrate potential applications of their results in other fields such as neuromorphic computing. These results, although preliminary, are intriguing. Overall, I find this a very interesting study regarding the biological components in SNN modeling for increasing the performance in complex temporal computing tasks. Yet, I have some concerns, especially with respect to the advance from prior work regarding the fusion mechanism and the statistics. Following are my major and minor comments.

Major:

1. Equation (3): The definition of the alpha timing factor given is erroneous; it follows from equation (2) that alpha is $1-1/\tau$. As the authors did not use the tau but alpha for learning, this will most probably not change the results, but please correct and indicate if this is the case.
2. Fig. 2b. This is a known function of the dendrites, termed multiplexing, as shown by Naud, Sprekeler PNAS 2018. Is the necessity for fusion different from this prior work, or does it follow the same principle?
3. All results presented in this work are from one network instance. Given the random initialization, repetitions, in order to derive appropriate statistics, are missing (error bars, significance). This is also important for the interpretation of results, as, for example, in Fig. 5f, where there is a non-intuitive decrease in performance for L3, vanilla SNN.

4. The authors make a very interesting argument for the necessity of the fusion of temporal features to achieve high performance. In the case of the 2 dendrites in the multi-timescale XOR problem, is the spatial restriction of signals 1 and 2 to the specific branches necessary? It would be interesting to show how the alpha parameter would evolve during training in the case of mixed inputs, e.g., do the dendrites acquire the imposed selectivity by randomly adjusting their temporal factor?

5. Please add and compare the results of the SHD and SSC with the ones of Rossbroich et al. 2022.

6. Fig 7. As it is argued that temporal heterogeneity (implemented via the dendrites) is needed to account for the intrinsic multi-timescale components in EEG signals, the authors should compare the results of the DH-SFNNs with the SRNNs/ DH-SRNNs. The same applies for Sup. Fig. 12.

7. What are the synaptic dynamics (I_{ext}) incorporated in the model and how do the authors compare a `vanilla SNN` with synaptic current dynamics to the DH-LIF model? This is of high importance, and it pertains to the model's novelty and the appropriateness of comparisons made in Table 1.

Minor

Equation (3): What are the assumptions for Δt or what was the actual Δt used?

Fig. 2a: Please indicate in the Fig. legend what the red box is highlighting.

Fig. 3b. Please indicate clearly in the Fig. legend that the difference between the 3 vanilla SFNNs is in the initialization of the beta parameter.

Fig. 3d: Here and in other similar figures, please indicate what the solid lines indicate.

Fig. 4a. Please add a detailed description of the figure in the legend. What is shown on the left and what is on the right panel? (the same applies also for figs 4d, and 4e. In the latter case, please indicate if the upper panels correspond to the small initializations, etc.). What does the "slow" and "fast" mean, is it not Signal 1 Signal 2, as in the left panel?

Fi. 4b. Please indicate in Fig. Legend what is a beneficial initialization. Even though this is explained in the main text, it should be also stated in the Fig. to allow for a better understanding of what is plotted.

Fig. 4e. As there is a single dendritic current per neuron in the one-branch DH-LIF neurons, how did the authors distinguish between the dendritic current responses to signal 1 and signal 2?

Fig. 3f and 4g grey and red bars show the same results (for the large initialization of timing factors and learnable parameters). For ease of reading, please consider merging these results for SHD and SSC in one figure. The same applies to Supp Fig 4 and Fig 4f, which are not referenced together in the text.

Fig. 3g. Please change the position of the labels, as it is, it obscures the data.

Fig. 4g. Indicate how signals 1 and 2 are located at the dendrites when increasing the number of the dendrites.

Fig. S5. What is the reason for degrading accuracy with increasing the number of dendrites?

Fig. 5. Type 2 neuron, 1st layer of the 2-layer SFNN, right panel. The highlighted neurons respond persistently for both signals 1 and 2, not supporting the statement that “Type 2 neuron represents the neuron sensitive to Signal 1 with low frequency” (as it is evident from the 1-layer SRNN Type-2 neurons below).

Fig 5e,f & Fig 6c. Please use the same colormaps for consistency (all of which plot show accuracy).

Fig 5. “Given above analyses, it looks clear that the inter-branch feature fusion in a neuron, the inter-neuron feature fusion in a recurrent layer, and the inter-layer feature fusion in a network have similar and synergetic effects in capturing the multi-timescale temporal features, which are beneficial for performing multi-timescale temporal computing tasks.”. This is a very interesting result, and it would be nice to highlight that in Fig. 4d Type 3 neurons are present (maybe rename them to be consistent throughout the figures?)

Fig. 6b. “As predicted, there is no significant increase of parameters and operations as the number of dendritic branches.” As no statistical measurements were made, please remove the significant statement.

Fig. 6b. Why is there an increase of synaptic accumulations for 4 dendrites, followed by a drop for 8?

What is the accuracy threshold to consider a good performance throughout this work? (e.g., compare results in the red box of Supp Fig. 9 and 4b- one branch DH-SFNN that is stated that it fails in performing the multi-timescale spiking XOR problem).

Please report if a grid search has been done for the optimal parameters and why in some datasets the authors use validation sets while in others no.

Reviewer #2 (Remarks to the Author):

In this work, the authors proposed a multi-compartment spiking neural network model with temporal dendritic heterogeneity, as well as a training mechanism for that approach.

There have been several recent works in the field on the inclusion of multi-compartment spiking neural networks with dendrites. I would like to have seen discussion of these approaches and how this approach compares to those in terms of results. These works include:

Yang, Shuangming, Tian Gao, Jiang Wang, Bin Deng, Benjamin Lansdell, and Bernabe Linares-Barranco. "Efficient spike-driven learning with dendritic event-based processing." *Frontiers in Neuroscience* 15 (2021): 601109.

Gao, Tian, Bin Deng, Jiang Wang, and Guosheng Yi. "Highly efficient neuromorphic learning system of spiking neural network with multi-compartment leaky integrate-and-fire neurons." *Frontiers in Neuroscience* 16 (2022): 929644.

It is worth noting that these works do not provide as comprehensive a comparison on existing datasets as this work, but it would still have been nice to see a discussion of how this approach compares.

I appreciated the inclusion of the temporal XOR experiment (in addition to the known, real-world datasets) to demonstrate the effectiveness of the inclusion of different time scales.

I have several questions/concerns with the experimental setup/results:

1) It is noted in the work that dendritic branches can approximate synaptic heterogeneity and there are results in the supplementary figures that show saturation. It would have been nice to see a direct comparison with approaches that leverage heterogeneous synapses, especially to strengthen the argument that heterogeneous dendrites are more computationally efficient.

2) The architectural decisions, particularly with respect to how the dendrites are connected to the inputs, appear to be fixed across the experiments. It would have been nice to see what impact different types of connectivity have, and whether having heterogeneous number of dendritic connections across neurons in the network is worthwhile.

3) It is noted that accuracy score tend to increase as the number of layers grow, but that there is evidence that the performance will become saturated. I would like to have seen discussions as to the impact on the BPTT-like algorithm on increasing the number of layers, particularly with respect to learning the temporal parameters of the dendrites, and whether there are fundamental limitations in network depth for this algorithmic approach.

4) For non-spiking datasets that were evaluated, how was that data encoded into spikes for the spiking neural network approaches? I would imagine that the encoding scheme chosen can have great impact on accuracy.

Overall, the methodology is sound. I appreciated the diversity in experiments and datasets evaluated, but as noted above, there are some components that I would like to have seen more exploration of. Once the code is made available, there appear to be enough details to reproduce the work for most cases. However, there was a lack of description about how to encode the data for non-spiking sets that I would like to have seen more detail on.

This is a compelling work with very interesting results. My main concerns are that I would like to have seen it placed in more context in the spiking neural network/neuromorphic field in terms of other works that are investigating the impact of dendritic computation on performance, and that there are several components of the approach that I do not feel like were fully explored in this work.

Reviewer #3 (Remarks to the Author):

Comments on "Temporal Dendritic Heterogeneity Incorporated with Spiking Neural Networks for Learning Multi-timescale Dynamics":

In this manuscript, Zheng and colleagues investigate the role of dendritic heterogeneity, specifically in integration time scales, within a spiking neuronal model (leaky integrate and fire). Their study aims to determine if incorporating dendritic heterogeneity can improve performance in tasks that require temporal integration. Overall, the results presented support the hypothesis that including dendritic heterogeneity is beneficial in terms of performance. These findings are valuable to the machine learning community, as they demonstrate how incorporating biological features can enhance model performance. However, the manuscript requires improvements in language, illustrations, and the interpretation of results. With significant enhancements, this manuscript has the potential to be suitable for publication in the Nature Communications journal.

Below are comments on various aspects of the manuscript (not listed in order of importance or appearance in the text):

1. I am curious if all the models compared in Figure 3 have the same number of parameters. Additionally, it seems intuitive that the addition of a learnable parameter would increase model accuracy. Furthermore, the vanilla SNN with a learnable somatic α parameter is missing from the comparisons. It would be expected that the vanilla SNN with an additional learnable parameter would show improved performance. Since the manuscript focuses on dendrites, I suggest including the vanilla SNN + learnable somatic α in Figures 3 and 4.

2. All presented results lack standard deviation, which leads me to assume that each network was initialized only once per case. However, given the inherent randomness in both artificial neural networks (ANNs) and spiking neural networks (SNNs), it's important to demonstrate robustness by reporting not only the best performing model but also the results across multiple initializations. I recommend running the experiments with at least 10 initializations per network and reporting average accuracies with standard deviations (e.g., Figure 4g).

3. It is crucial to clarify in the text that the DH-LIF model presented here is not a two-compartment model. Instead, it consists of a soma compartment and a variable number of dendritic branches. This clarification is important as it affects the number of trainable parameters, with each branch having its own α parameter.

4. Figure 2 appears to be confusing. It is not immediately evident why the dendritic time constant should match the input frequency. Multi-timescale dynamics arise from various dendritic and synaptic mechanisms, such as ionic channels and NMDARs. I suggest explaining the role of the dendritic time constant in this context and why it is unbounded (within biological ranges). A time scale of approximately 100ms is primarily influenced by NMDA synaptic characteristics rather than internal membrane properties. For instance, in the Methods section, the dendritic time constant is a function of membrane capacitance and resistances, which are structural properties of the dendrite and should be bounded within a biological regime.

5. Writing quality: While I acknowledge the effort put into creating the illustrations and writing the text, the manuscript would benefit from improved language. There are several grammatical and semantic issues that could be easily resolved using grammar-checking software. Additionally, all figure legends should be more descriptive and self-explanatory. Furthermore, the authors employ terminology that can confuse readers (e.g., agile capability, feature fusion, etc.).

Response Letter

Manuscript ID: NCOMMS-23-35049

Title: Temporal Dendritic Heterogeneity Incorporated with Spiking Neural Networks for Learning Multi-timescale Dynamics

Authors: Hanle Zheng, Zhong Zheng, Rui Hu, Bo Xiao, Yujie Wu, Fangwen Yu, Xue Liu, Guoqi Li, Lei Deng

We would like to thank the reviewers for spending valuable time and raising insightful comments which we feel have substantially improved our manuscript. The point-by-point responses are provided as follows. All the revisions in the manuscript and Supplementary Information (SI) are marked in blue.

1. Responses to Reviewer 1

Overall Comment: *This is a very interesting study by Zheng et al., where they investigate how heterogeneity, introduced by incorporating dendrites with different temporal scales, can be exploited for the processing of temporally variable signals. This work builds on the previous work of Perez-Nieves, et al. Nat. Communications 2021, and shows that spiking neural network (SNNs) models with temporal dendritic heterogeneity enable multi-timescale dynamics by learning heterogeneous timing factors on different dendritic branches. Specifically, the authors report higher accuracy, robustness, and generalization compared to vanilla SNNs. Importantly, they show that a fusion of features (similar to the multiplexing of Naud, Sprekeler PNAS 2018) is needed for performing multi-timescale tasks and that this fusion can be achieved either within a single neuron, between neurons in a recurrent layer, or between layers. This necessity further indicates that high performance in multi-timescale tasks cannot be simply achieved by increasing the number of parameters within a layer. The authors further seek to demonstrate potential applications of their results in other fields such as neuromorphic computing. These results, although preliminary, are intriguing. Overall, I find this a very interesting study regarding the biological components in SNN modeling for increasing the performance in complex temporal computing tasks. Yet, I have some concerns, especially with respect to the advance from prior work regarding the fusion mechanism and the statistics. Following are my major and minor comments.*

Response: We greatly appreciate the reviewer giving such positive feedback and insightful comments. We have carefully considered your concerns, provided detailed responses, and revised the manuscript accordingly. We hope our following responses and the revised manuscript can make you satisfied.

Major Comment 1: *Equation (3): The definition of the alpha timing factor given is erroneous; it follows from equation (2) that alpha is 1-1/tau. As the authors did not use the tau but alpha for learning, this will most probably not change the results, but please correct and indicate if this is the case.*

Response: In Equation (2), the time constant τ_d follows

$$\tau_d \frac{\partial i_d^t}{\partial t} = -i_d^t + I_d^t. \quad (1)$$

We discretize the above equation with the Euler method, which can be written as

$$\tau_d(i_d^t - i_d^{t-1}) = -i_d^t + I_d^t. \quad (2)$$

Then we can have

$$i_d^t = \left(1 - \frac{1}{1 + \tau_d}\right)i_d^{t-1} + \frac{1}{1 + \tau_d}I_d^t, \quad (3)$$

which is equivalent to

$$i_d^{t+1} = \left(1 - \frac{1}{1 + \tau_d}\right)i_d^t + \frac{1}{1 + \tau_d}I_d^{t+1}. \quad (4)$$

Finally, we denote $\alpha_d = 1 - \frac{1}{1 + \tau_d}$ and yield

$$i_d^{t+1} = \alpha_d i_d^t + (1 - \alpha_d)I_d^{t+1}. \quad (5)$$

Certainly, in another situation, if we discretize $\tau_d \frac{\partial i_d^t}{\partial t} = -i_d^t + I_d^t$ like

$$\tau_d(i_d^{t+1} - i_d^t) = -i_d^t + I_d^t, \quad (6)$$

we can similarly have

$$i_d^{t+1} = \left(1 - \frac{1}{\tau_d}\right)i_d^t + \frac{1}{\tau_d}I_d^t. \quad (7)$$

Finally, we denote $\alpha_d = 1 - \frac{1}{\tau_d}$ and yield

$$i_d^{t+1} = \alpha_d i_d^t + (1 - \alpha_d)I_d^{t+1}. \quad (8)$$

Although the above two formats have different definitions of α_d with respect to τ_d , they share a unified representation if we only look at the α_d level. As the reviewer pointed out, it will not change the results not matter we define $\alpha_d = 1 - \frac{1}{1 + \tau_d}$ or $\alpha_d = 1 - \frac{1}{\tau_d}$ if the learnable parameter is α_d rather than τ_d . In our implementation, we define $\alpha_d = 1 - \frac{1}{1 + \tau_d}$.

Revision: Added clarifications with Equation (3)&(4) in Methods.

Major Comment 2: *Fig. 2b. This is a known function of the dendrites, termed multiplexing, as shown by Naud, Sprekeler PNAS 2018. Is the necessity for fusion different from this prior work, or does it follow the same principle?*

Response: Thanks for the reviewer's kind reminder. In essence, the prior work (Naud, Sprekeler, PNAS 2018 [1]) had been cited as ref. 29 in our previous manuscript. It is correct that the multiplexing is a known function of dendrites in computational neuroscience. The authors of that prior work proposed a type of multiplexing based on separation of bursts and single spikes at the level of an ensemble, allowing neurons to represent multiple information streams simultaneously without ambiguity.

Although both of the prior work and our work mention the temporal fusion function of dendrites, the motivations are different. We mainly focus on the effectiveness of the proposed model inspired by the biological observations for solving complex temporal computing tasks in practice. Therefore, we must consider how it can benefit spiking neural networks in those tasks with acceptable computational complexity and effective learning algorithms. Differently, the prior work tried to understand how the hierarchical communication in the brain where bottom-up and top-down information must be distinguished could be implemented with multiplexing dendrites. Their experiments were conducted on simulated biological neural models without considering how the mechanism could benefit the solving of complex

temporal computing tasks. With the above analyses, we think it is necessary to alter the fusion mechanism for making it applicable to practical tasks.

It is worth noting that the principles followed by the two works also have similarities. For example, the neuron model itself must incorporate multi-timescale temporal dynamics to fit multi-timescale information flows for processing complex temporal signals. In the prior work, the authors mentioned that they simulated the response of an ensemble of hick-tufted pyramidal neurons (TPNs) receiving two independent input signals with different frequencies: one injected into dendrites and the other injected in to the soma. They further quantified the encoding quality in multiplexing at different timescales by calculating the frequency-resolved coherence between the inputs and the estimates. They found that the coherence between the dendritic inputs and the estimates based on the burst probability is close to one for slow input fluctuations, but decreases to zero for rapid input fluctuations, which is similar to our dendritic branch modeling with large timing factors. In the meantime, they found that the event rate can decode the soma input with high accuracy for input frequencies up to 100 Hz, which is similar to our dendritic branch modeling with small timing factors.

Revision: Added more texts when describing Fig. 2b to clarify the similarities and differences compared to the prior work.

Major Comment 3: *All results presented in this work are from one network instance. Given the random initialization, repetitions, in order to derive appropriate statistics, are missing (error bars, significance). This is also important for the interpretation of results, as, for example, in Fig. 5f, where there is a non-intuitive decrease in performance for L3, vanilla SNN.*

Response: This is a nice suggestion. In this work, we conducted many experiments in several complex temporal tasks, which are very time-consuming and make the repetitions challenging. To fix this problem raised by the reviewer with acceptable time overheads, we run the experiments with 10 random initializations in the customized spiking XOR tasks and 5 random initializations on SHD and SCC datasets, and report average accuracies with standard deviations in the revised manuscript. With updated results, our conclusions still hold. Especially, there is no non-intuitive decrease in performance for L3, vanilla SNN (Fig. 5f) in the revised version.

Revision: Added repeated experiments and updated the reported results.

Major Comment 4: *The authors make a very interesting argument for the necessity of the fusion of temporal features to achieve high performance. In the case of the 2 dendrites in the multi-timescale XOR problem, is the spatial restriction of signals 1 and 2 to the specific branches necessary? It would be interesting to show how the alpha parameter would evolve during training in the case of mixed inputs, e.g., do the dendrites acquire the imposed selectivity by randomly adjusting their temporal factor?*

Response: We restrict Signal 1 and Signal 2 connecting to two different branches mainly for the purpose of clear presentation. In fact, we have also conducted experiments in which the spatial restriction is removed and the synapses with Signal 1 and Signal 2 are randomly connected to the dendritic branches. The results keep unchanged, which would be found in the added Supplementary Fig. S4a.

The evolving of α parameters in the case of mixed inputs has also been visualized in the added Supplementary Fig. S4b-c. It can be seen that the distributions of α gradually exhibit three peaks different from the initialization, which evidences that the learning process can make the dendritic timing factors acquire selectivity to multiple timescales of input signals.

Revision: Added Supplementary Fig. S4 to show the case of mixed inputs.

Major Comment 5: *Please add and compare the results of the SHD and SSC with the ones of Rossbroich et al. 2022.*

Response: Please note that in the mentioned work (Rossbroich et al. 2022 [2], ref. 38 in the revised manuscript), they only show the results on the SHD dataset rather than the SSC dataset. In the revised Table 1, we have added comparison with their results on SHD. It can be seen that our model can achieve higher accuracy with fewer parameters.

Revision: Added comparison with Rossbroich et al. 2022 (ref. 38) in the revised Table 1.

Major Comment 6: *Fig 7. As it is argued that temporal heterogeneity (implemented via the dendrites) is needed to account for the intrinsic multi-timescale components in EEG signals, the authors should compare the results of the DH-SFNNs with the SRNNs/DH-SRNNs. The same applies for Sup. Fig. 12.*

Response: According to the reviewer’s suggestion, we have added extra experiments of vanilla SRNNs and DH-SRNNs in the EEG recognition and robot place recognition tasks in the revised Supplementary Fig. S12&S13. It can be seen that the temporal heterogeneity still helps improve the performance. Moreover, compared to the results with DH-SFNNs in the EEG recognition task, the accuracy gap presents a reduced trend between DH-SRNNs when varying the number of dendritic branches, which evidences again the faster performance saturation of SRNNs.

Revision: Added extra experiments of vanilla SRNNs and DH-SRNNs in Supplementary Fig. S12&S13.

Major Comment 7: *What are the synaptic dynamics (I_{ext}) incorporated in the model and how do the authors compare a ‘vanilla SNN’ with synaptic current dynamics to the DH-LIF model? This is of high importance, and it pertains to the model’s novelty and the appropriateness of comparisons made in Table 1*

Response: In our models, I_{ext} is the weighted integration of spiking inputs from pre-synaptic neurons. We did not introduce synaptic dynamics into our models and the vanilla SNNs to avoid the influence of synaptic dynamics and focus on analyzing the dendritic dynamics.

In fact, the LIF neuron with synaptic dynamics has been widely studied in the field of SNNs. For example, in ref. 31 (Perez-Nieves, et al. Nature Communications 2021 [3]), the LIF neuron is modelled as

$$\tau_m \frac{\partial U_i^t}{\partial t} = -(U_i^t - U_0) + I_i^t, \quad (9)$$

$$\tau_s \frac{\partial I_i^t}{\partial t} = -I_i^t + I_{ext}^t, \quad (10)$$

where U_i^t and I_i^t represent the membrane potential and the synaptic current of the i -th neuron, τ_m and τ_s denote the time constants for the membrane potential and the synaptic current, respectively. This model is quite similar to our DH-LIF neuron with a single dendritic branch in each neuron, i.e., the synaptic current dynamics here equivalent to dendritic dynamics of the single dendritic branch.

With above knowledge, it is clear that our results have included the vanilla SNNs with synaptic current dynamics if we look at the results of DH-SNNs with one dendritic branch. It is worth noting that vanilla SNNs with synaptic current dynamics cannot generate temporal dendritic heterogeneity by fusing multi-timescale temporal features at the neuron level because there is only a single dendritic branch in each

neuron. Whereas, the temporal dendritic heterogeneity, proposed in this work, is very important for solving complex temporal computing tasks, which has been evidenced by our results of DH-SNNs with multiple dendritic branches in each neuron.

Revision: In the second paragraph of the “*Long-term memory via dendritic dynamics*” part of the revised manuscript, explicitly clarified the equivalence of vanilla SNNs with synaptic current dynamics and our one-dendritic-branch DH-SNNs.

Minor Comment 1: *Equation (3): What are the assumptions for δt or what was the actual δt used?*

Response: The actual δt depends on what the time resolution used in different tasks. In most cases, the assumed δt is $1ms$. While in Fig. 3g, when we compare the recognition accuracy of vanilla SFNNs and one-dendritic-branch DH-SFNNs with learnable timing factors under different sampling time intervals on SHD and SSC, the δt is set to $1ms$, $2ms$, $4ms$ or $8ms$.

Minor Comment 2: *Fig. 2a: Please indicate in the Fig. legend what the red box is highlighting.*

Response: The red box is to highlight that each dendritic branch has a temporal memory on the dendritic current with a variable timing factor.

Revision: Modified the legend of Fig. 2a.

Minor Comment 3: *Fig. 3b. Please indicate clearly in the Fig. legend that the difference between the 3 vanilla SFNNs is in the initialization of the beta parameter.*

Response: The three different initializations of β can be found in Supplementary Table S3, which follow three uniform distributions of $U(-4, 0)$ (small), $U(0, 4)$ (medium), and $U(2, 6)$ (large).

Revision: Modified the legend of Fig. 3b.

Minor Comment 4: *Fig. 3d: Here and in other similar figures, please indicate what the solid lines indicate.*

Response: Each solid line is a kernel density estimating (KDE) curve of the corresponding histogram.

Revision: Modified the legends of Fig. 3d and Fig. 4c.

Minor Comment 5: *Fig. 4a. Please add a detailed description of the figure in the legend. What is shown on the left and what is on the right panel? (the same applies also for figs 4d, and 4e. In the latter case, please indicate if the upper panels correspond to the small initializations, etc.). What does the “slow” and “fast” mean, is it not Signal 1 Signal 2, as in the left panel?*

Response: The left and right panels of Fig. 4a represent two independent input examples, and the left and right panels of Fig. 4d and 4e accordingly visualize the network states for the corresponding input example. In Fig. 4e, the upper and lower panels correspond to the network states under the small and large initialization of β , respectively. Here the “slow” and “fast” indicate the frequency (or timescale) of the input signals. Specifically, Signal 1 is a slow signal with low frequency that presents a long timescale, i.e., changing slowly, while Signal 2 is a fast signal with high frequency that presents a short timescale, i.e., changing rapidly.

Revision: Modified the legends of Fig. 4a/d/e and Fig. 5a.

Minor Comment 6: *Please indicate in Fig. Legend what is a beneficial initialization. Even though this is explained in the main text, it should be also stated in the Fig. to allow for a better understanding of what is plotted.*

Response: In the previous manuscript, the beneficial initialization is explained in the figure captions. According to this comment, we have modified the figure legends to make the definition clearer.

Revision: Modified the legend of Fig. 4d.

Minor Comment 7: *Fig. 4e. As there is a single dendritic current per neuron in the one-branch DH-LIF neurons, how did the authors distinguish between the dendritic current responses to signal 1 and signal 2?*

Response: Yes, there is a single dendritic current per neuron in the one-dendritic-branch DH-LIF neurons. During training, all input signals of a neuron are connected to the single dendritic branch. After training, we distinguish the responses by separately inputting Signal 1 or Signal 2. For example, when we show the response of Signal 1, we just inject Signal 1 to DH-LIF neurons and observe the dendritic current responses.

Revision: Added testing details in the legend of Fig. 4e.

Minor Comment 9: *Fig. 3f and 4g grey and red bars show the same results (for the large initialization of timing factors and learnable parameters). For ease of reading, please consider merging these results for SHD and SSC in one figure. The same applies to Supp Fig 4 and Fig 4f, which are not referenced together in the text.*

Response: Yes, a small part of the results in Fig. 3f and Fig. 4g (now Fig. 4f in the revised manuscript) on SHD and SSC datasets are repeated. However, at a higher level, Fig. 3 and Fig. 4 talk different story lines. Fig. 3 presents the results for one-dendritic-branch DH-SNNs, while Fig. 4 presents the results for multi-dendritic-branch DH-SNNs (including one-dendritic-branch DH-SNNs) to observe the influence of variable numbers of dendritic branches. Although merging the results on SHD and SSC datasets is a nice advice, we decide not to merge considering the distinct motivations of the whole Fig. 3 and Fig. 4. We hope you can understand our concern.

For Supplementary Fig. S4 and Fig. 4f in the previous manuscript, we have merged them to form an updated Supplementary Fig. S5 and referenced them together in the last paragraph of the “*Long-term memory via dendritic dynamics*” part of the revised manuscript according to your advice.

Revision: Updated Supplementary Fig. S5 by merging Fig. 4f and Supplementary Fig. S4 in the previous manuscript, and then modified the figure reference.

Minor Comment 10: *Fig. 3g. Please change the position of the labels, as it is, it obscures the data.*

Response: We guess you mean Fig. 4g (now Fig. 4f in the revised manuscript) rather than Fig. 3g. We have changed the positions of the labels in Fig. 4b/f.

Revision: Changed label positions of Fig. 4b/f.

Minor Comment 11: *Fig. 4g. Indicate how signals 1 and 2 are located at the dendrites when increasing the number of the dendrites.*

Response: Please note that Fig. 4g (now Fig. 4f in the revised manuscript) shows the results in SHD and SSC tasks other than the multi-timescale spiking XOR tasks, so there are no signals 1 and 2 in this subfigure.

In fact, we had clarified how the pre-synaptic inputs connect to the dendritic branches in the beginning of the “*Spiking neural network with DH-LIF neurons (DH-SNN)*” part of the previous manuscript: “*In order to avoid the parameter exploding as the number of dendritic branches grows, we add a sparse restriction on the connection pattern between neurons (see Methods - Part: Sparse connection restriction). For each neuron, the pre-synaptic inputs are randomly distributed on the dendritic branches. The sets of input indexes on different branches are non-overlapped and the number of inputs keeps identical across branches to the greatest extent*”.

Minor Comment 12: *Fig. S5. What is the reason for degrading accuracy with increasing the number of dendrites?*

Response: Notice that, in the revised manuscript, Supplementary Fig. S5 becomes Supplementary Fig. S6. Due to the sparse connection restriction we added, the sparsity ratio decreases as the number of dendritic branches grows, which might degrade the accuracy. Moreover, the model needs to learn more dendritic timing factors with more dendritic branches, which instead complicates the training process if the task does not need pretty much temporal dendritic heterogeneity. Therefore, in our work, we just claim that the accuracy can be improved as the number of dendritic branches properly grows, while can be saturated and even degraded when it exceedingly grows. The performance degradation is an interesting phenomenon and we would like to conduct in-depth investigation in the future.

Minor Comment 13: *Fig. 5. Type 2 neuron, 1st layer of the 2-layer SFNN, right panel. The highlighted neurons respond persistently for both signals 1 and 2, not supporting the statement that “Type 2 neuron represents the neuron sensitive to Signal 1 with low frequency” (as it is evident from the 1-layer SRNN Type-2 neurons below).*

Response: For Type 2 neurons in the first hidden layer of the two-layer SFNN, we give the statement that “*The Type 2 neuron represents the neuron sensitive to Signal 1 with low frequency*” by comparing the same neurons’ responses in different input cases as the left and right panels present.

In the left panel, when Signal 1 with a low firing rate is inputted, Type 2 neurons exhibit sparse spiking activities. On the contrary, in the right panel, when Signal 1 with a high firing rate is inputted, Type 2 neurons display dense spiking activities. Furthermore, the responses of Type 2 neurons are uniformly distributed, not influenced by the periodically changing Signal 2. With these observations, we conclude that Type 2 neurons are sensitive to Signal 1.

Revision: Added explanations for the selectivity of highlighted neurons when referencing Fig. 5a.

Minor Comment 14: *Fig 5e,f & Fig 6c. Please use the same colormaps for consistency (all of which plot show accuracy).*

Response: Thank you for carefully pointing out this problem. The same colormaps have been applied to Fig. 5e,f and Fig. 6c, respectively. Given the big gap between the accuracy ranges in Fig. 5 and Fig. 6, we use different colormap ranges for them to avoid unclear presentation.

Revision: Unified the colormaps in Fig. 5e,f and Fig. 6c, respectively.

Minor Comment 15: *Fig 5. “Given above analyses, it looks clear that the inter-branch feature fusion*

in a neuron, the inter-neuron feature fusion in a recurrent layer, and the inter-layer feature fusion in a network have similar and synergetic effects in capturing the multi-timescale temporal features, which are beneficial for performing multi-timescale temporal computing tasks.” This is a very interesting result, and it would be nice to highlight that in Fig. 4d Type 3 neurons are present (maybe rename them to be consistent throughout the figures?)

Response: According to your suggestion, we have renamed them as “*Highlighted neurons*” for consistency.

Revision: Unified the names of highlighted neurons in Fig. 4d and Fig. 5a.

Minor Comment 16: *Fig. 6b. “As predicted, there is no significant increase of parameters and operations as the number of dendritic branches grows.” As no statistical measurements were made, please remove the significant statement.*

Response: We have removed the significant statement when referencing Fig. 6b in the revised manuscript.

Revision: Removed the significant statement.

Minor Comment 17: *Fig. 6b. Why is there an increase of synaptic accumulations for 4 dendrites, followed by a drop for 8?*

Response: For an SNN layer, the number of synaptic accumulations depends on both the synaptic connections and the input firing rates. For example, assuming that an SRNN layer has N neurons with M inputs, the number of synaptic accumulations per timestep can be written as

$$N_{syn_ac} = MNr_{in} + N^2r_{out} \quad (11)$$

where the r_{in} and r_{out} represent the mean firing rates of spike inputs and outputs, respectively. The above equation can be found in Fig. 6a. Although our sparse connection restriction remains the number of synaptic connections unchanged to a great extent as the number of dendritic branches grows, the actual number of synaptic accumulations can fluctuate due to variable firing rates of spiking activities, which had been explained in the previous manuscript when referencing Fig. 6b.

Minor Comment 18: *What is the accuracy threshold to consider a good performance throughout this work? (e.g., compare results in the red box of Supp Fig. 9 and 4b- one branch DH-SFNN that is stated that it fails in performing the multi-timescale spiking XOR problem).*

Response: We apologize for the unclear definition of accuracy thresholds. First, please note that different tasks with variable difficulty should have different accuracy thresholds. In the context of the multi-timescale spiking XOR problem in Fig. 4b, we place the accuracy threshold at 75% which is acceptable, although not so rigorous, for a typical XOR problem processed by a model without complex feature fusion. However, for the simple SHD task, the defined “well-performing” networks should achieve accuracy surpassing 88%; while for the difficult SSC task, the accuracy threshold decreases to 69%. We know the latter two thresholds look a little specific, but our purpose is to restrict the size of the red box and explicitly present the accuracy degradation phenomenon under too small or too large sparsity levels, which is acceptable if we look at the whole Supplementary Fig. S9 (now is Supplementary Fig. S10 in the revised manuscript).

Revision: Clarified the accuracy thresholds when describing the failing performance.

Minor Comment 19: *Please report if a grid search has been done for the optimal parameters and why in some datasets the authors use validation sets while in others no.*

Response: The grid search was not adopted in selecting the optimal parameters in our study. Instead, we adopted parameter settings similar to those utilized in a prior work (Yin et al., Nat. Mach. Intell., 2021 [4], ref. 33 in the manuscript). Specifically, the hyper-parameters such as the parameters used in the surrogate gradient function, the learning rate and the batch size are simply set to make the training successful. The setting of the timing factors is based on the experiences in preliminary experiments on the SSC dataset. All parameter settings keep unchanged in experiments on the same dataset to ensure the reliability of the ablation study results. We believe the reviewer points out a nice topic on the search of optimal parameters and we leave it for future work. In addition, we maintained consistency by employing the same validation set configuration also as in the mentioned prior work.

REFERENCES

- [1] R. Naud and H. Sprekeler, "Sparse bursts optimize information transmission in a multiplexed neural code," *Proceedings of the National Academy of Sciences*, vol. 115, no. 27, pp. E6329–E6338, 2018.
- [2] J. Rossbroich, J. Gygax, and F. Zenke, "Fluctuation-driven initialization for spiking neural network training," *Neuromorphic Computing and Engineering*, vol. 2, no. 4, p. 044016, 2022.
- [3] N. Perez-Nieves, V. C. Leung, P. L. Dragotti, and D. F. Goodman, "Neural heterogeneity promotes robust learning," *Nature communications*, vol. 12, no. 1, pp. 1–9, 2021.
- [4] B. Yin, F. Corradi, and S. M. Bohté, "Accurate and efficient time-domain classification with adaptive spiking recurrent neural networks," *Nature Machine Intelligence*, vol. 3, no. 10, pp. 905–913, 2021.

2. Response to Reviewer 2

Overall Comment: *In this work, the authors proposed a multi-compartment spiking neural network model with temporal dendritic heterogeneity, as well as a training mechanism for that approach.*

There have been several recent works in the field on the inclusion of multi-compartment spiking neural networks with dendrites. I would like to have seen discussion of these approaches and how this approach compares to those in terms of results. These works include:

*Yang, Shuangming, Tian Gao, Jiang Wang, Bin Deng, Benjamin Lansdell, and Bernabe Linares-Barranco. "Efficient spike-driven learning with dendritic event-based processing." *Frontiers in Neuroscience* 15 (2021): 601109.*

*Gao, Tian, Bin Deng, Jiang Wang, and Guosheng Yi. "Highly efficient neuromorphic learning system of spiking neural network with multi-compartment leaky integrate-and-fire neurons." *Frontiers in Neuroscience* 16 (2022): 929644.*

It is worth noting that these works do not provide as comprehensive a comparison on existing datasets as this work, but it would still have been nice to see a discussion of how this approach compares.

I appreciated the inclusion of the temporal XOR experiment (in addition to the known, real-world datasets) to demonstrate the effectiveness of the inclusion of different time scales.

Overall, the methodology is sound. I appreciated the diversity in experiments and datasets evaluated, but as noted above, there are some components that I would like to have seen more exploration of. Once the code is made available, there appear to be enough details to reproduce the work for most cases. However, there was a lack of description about how to encode the data for non-spiking sets that I would like to have seen more detail on.

This is a compelling work with very interesting results. My main concerns are that I would like to have seen it placed in more context in the spiking neural network/neuromorphic field in terms of other works that are investigating the impact of dendritic computation on performance, and that there are several components of the approach that I do not feel like were fully explored in this work.

Response: We really appreciate the reviewer for the generous praises of our work. For the concerns you mentioned above, we have made the following revisions including comparing the approaches and results of the mentioned references with our work, adding explanations on how to encode non-spiking data, and adding discussions on exploring more dendritic functions. Details can be found in the responses below, which we hope can address your concerns.

Comment 1: *It is noted in the work that dendritic branches can approximate synaptic heterogeneity and there are results in the supplementary figures that show saturation. It would have been nice to see a direct comparison with approaches that leverage heterogeneous synapses, especially to strengthen the argument that heterogeneous dendrites are more computationally efficient.*

Response: In Supplementary Fig. S6, we have demonstrated that with the number of dendritic branches exceedingly grows, e.g., approaching synaptic heterogeneity with a variable timing factor for each synapse, the accuracy of DH-SNNs would saturate and even degrade. This saturation indicates properly increasing the number of dendritic branches is enough to learn multi-timescale dynamics for a specific task and usually performs better than directly applying synaptic heterogeneity with too many learnable timing

factors.

For the computational cost, it is clear that the dendritic heterogeneity can be more efficient than the synaptic heterogeneity. For example, assuming that a DH-SFNN layer has N neurons with M inputs, the number of learnable parameters equals

$$N_{pra} = MN + (2D + 1)N \quad (12)$$

as given in Fig. 6a, where D is the number of dendritic branches. According to our experiments, the accuracy would be saturated when the number of branches reaches 4 or 8, while there are hundreds of inputs in each layer. When we calculate the number of computational operations we can omit the influence of D since we usually have $D \ll M$.

In contrast, synaptic heterogeneity is a special case of dendritic heterogeneity when we increase D to M , and the number of learnable parameters becomes

$$N_{pra} = 3MN + N, \quad (13)$$

which is much larger than that in dendritic heterogeneity. In this case, we can no longer omit the influence of D when calculating the number of computational operations. Therefore, considering the little improvement and even degradation in accuracy when solving real-world temporal computing tasks if we apply synaptic heterogeneity, it is wiser to model dendritic heterogeneity like our DH-SNNs. We have added clarifications regarding the computational efficiency compared to synaptic heterogeneity in Supplementary Table S4.

Revision: Added descriptions on the comparison to synaptic heterogeneity in terms of computational efficiency in Supplementary Table S4.

Comment 2: *The architectural decisions, particularly with respect to how the dendrites are connected to the inputs, appear to be fixed across the experiments. It would have been nice to see what impact different types of connectivity have, and whether having heterogeneous number of dendritic connections across neurons in the network is worthwhile.*

Response: This is a very insightful comment. Actually, we had conducted experiments to make the connection pattern between dendrites and inputs learnable through the deep rewiring (DEEP R [1]) algorithm. The algorithm could automatically modify the network connections during training by pruning and rewiring synapses according to their significance. The experimental results of one-layer DH-SFNNs on SHD and SSC datasets are provided below.

TABLE I
RESULTS OF LEARNABLE CONNECTION PATTERN.

Datasets	Connection pattern	2-branch	4-branch	8-branch
SHD	Fixed	85.98%	88.15%	88.78%
	Learnable	86.20%	87.88%	88.60%
SSC	Fixed	68.49%	69.70%	70.17%
	Learnable	68.38%	69.62%	70.23%

We find it is hard to say whether the learnable connection pattern can benefit the performance of DH-SNNs or not. Maybe, the reason that performance cannot be obviously improved is due to the unstable learning process with a variable connection pattern. We had spent more than two months on this problem before the first submission, but still cannot get satisfactory results. Therefore, we did not put these results in the manuscript to avoid misleading readers and shifting our story line. We are pleased to share the

original experimental results with you, and of course we would like to conduct in-depth studies in future work.

Revision: Added discussions on learnable connections in the *Discussion* part of the revised manuscript.

Comment 3: *It is noted that accuracy score tend to increase as the number of layers grow, but that there is evidence that the performance will become saturated. I would like to have seen discussions as to the impact on the BPTT-like algorithm on increasing the number of layers, particularly with respect to learning the temporal parameters of the dendrites, and whether there are fundamental limitations in network depth for this algorithmic approach.*

Response: It is indeed essential to solve the well-known challenge in training deep neural networks caused by the gradient vanishing problem in the BP algorithm. A variety of strategies like the shortcut connection and normalization methods have been employed to alleviate the problem. In this work, DH-SNNs are also trained using a BPTT-like algorithm and the training of parameters also depends on the backpropagation of gradients, which of course suffers from the same issue.

In the *Discussion* part of the previous manuscript, we had pointed out that “*Because we focus on demonstrating the effectiveness of temporal dendritic heterogeneity and revealing its working mechanism in this work, we select the simple fully-connected rather than convolutional layers as the backbone and do not pay much attention on training optimization techniques*”. This is one of the reasons that we did not show improved performance as the networks deepen. It is valuable, as the reviewer pointed out, to introduce optimization techniques for improving the performance of deep DH-SNNs in future work.

Comment 4: *For non-spiking datasets that were evaluated, how was that data encoded into spikes for the spiking neural network approaches? I would imagine that the encoding scheme chosen can have great impact on accuracy.*

Response: For non-spiking datasets, the real-value inputs are directly fed into DH-SNNs. The first layer of DH-SNNs receives real-value inputs rather than spiking inputs, but still performs the spiking neural dynamics as a normal DH-SNN layer except for the different input format. In this way, the first layer actually acts as an encoding layer that converts non-spiking inputs to spiking outputs and then sends to post-synaptic layers. The parameters of the first layer are learned together with the network parameters for optimal performance. This is a widely used encoding method in the field of SNNs proposed in (Y. Wu, L. Deng, et al. 2019 [2]).

Revision: Added descriptions on the encoding scheme for non-spiking data in the “*Datasets and tasks*” part of *Methods*.

Comment 5: *My main concerns are that I would like to have seen it placed in more context in the spiking neural network/neuromorphic field in terms of other works that are investigating the impact of dendritic computation on performance, and that there are several components of the approach that I do not feel like were fully explored in this work.*

Response: There indeed exist some works that investigate the computational functions of dendrites in the fields of neuroscience, neuromorphic computing, and deep learning. For example, like the works mentioned by the reviewer (Yang et.al. 2021 [3]& Gao et.al. 2022 [4], cited as refs. 66-67 in the revised manuscript), the authors proposed an efficient spike-driven learning method based on dendritic computation and further implemented them on FPGA. In fact, the functions of biological dendrites are quite diverse such as local nonlinear transformation, adjustment to synaptic learning rules, multiplexing different sources of neural signals, and the generation of multi-timescale dynamics, making it difficult to incorporate all of them into

a single network.

In our work, we focus on incorporating the temporal dendritic heterogeneity mechanism into SNNs and training the networks to solve real-world temporal computing tasks using an advanced learning algorithm, and finally explaining how it works. Although many other computational functions of dendrites have not been fully explored, our work remains innovative and achieves state-of-the-art performance. We would like to continue exploring the remaining functions of dendrites in future work. Our goal is to solve practical problems by employing bio-inspired observations.

Revision: Added discussions on exploring more dendritic functions in the *Discussion* part of the revised manuscript.

REFERENCES

- [1] G. Bellec, D. Kappel, W. Maass, and R. Legenstein, "Deep rewiring: Training very sparse deep networks," *arXiv preprint arXiv:1711.05136*, 2017.
- [2] Y. Wu, L. Deng, G. Li, J. Zhu, Y. Xie, and L. Shi, "Direct training for spiking neural networks: Faster, larger, better," in *Proceedings of the AAAI Conference on Artificial Intelligence*, vol. 33, 2019, pp. 1311–1318.
- [3] S. Yang, T. Gao, J. Wang, B. Deng, B. Lansdell, and B. Linares-Barranco, "Efficient spike-driven learning with dendritic event-based processing," *Frontiers in Neuroscience*, vol. 15, p. 601109, 2021.
- [4] T. Gao, B. Deng, J. Wang, and G. Yi, "Highly efficient neuromorphic learning system of spiking neural network with multi-compartment leaky integrate-and-fire neurons," *Frontiers in Neuroscience*, vol. 16, p. 929644, 2022.

3. Response to Reviewer 3

Overall Comment: *In this manuscript, Zheng and colleagues investigate the role of dendritic heterogeneity, specifically in integration time scales, within a spiking neuronal model (leaky integrate and fire). Their study aims to determine if incorporating dendritic heterogeneity can improve performance in tasks that require temporal integration. Overall, the results presented support the hypothesis that including dendritic heterogeneity is beneficial in terms of performance. These findings are valuable to the machine learning community, as they demonstrate how incorporating biological features can enhance model performance. However, the manuscript requires improvements in language, illustrations, and the interpretation of results. With significant enhancements, this manuscript has the potential to be suitable for publication in the Nature Communications journal.*

Response: We sincerely thank the reviewer for the positive feedback on our motivation and performance benefits in this work. For the concerns about the language, illustrations, and the interpretation of results, we have tried our best to make revisions in the updated manuscript. Details can also be found in the following responses.

Comment 1: *I am curious if all the models compared in Figure 3 have the same number of parameters. Additionally, it seems intuitive that the addition of a learnable parameter would increase model accuracy. Furthermore, the vanilla SNN with a learnable somatic α parameter is missing from the comparisons. It would be expected that the vanilla SNN with an additional learnable parameter would show improved performance. Since the manuscript focuses on dendrites, I suggest including the vanilla SNN + learnable somatic α in Figures 3 and 4.*

Response: In Figure 3, we present a comparison of DH-SNNs equipped with a single dendritic branch and vanilla SNNs in terms of performance in the delayed XOR spiking task as well as real-world temporal tasks (SHD and SSC), with a focus on evaluating their long-term memory capability. As illustrated in Figure 6(a), the extra parameters introduced by DH-SNNs are at the level of individual neurons rather than the vast synapses. Specifically, the one-dendritic-branch DH-SNNs exhibit only 0.2% and 0.1% increases of parameters per network compared to vanilla SNNs on SHD and SSC datasets, respectively. Despite this marginal difference in parameters, the DH-SNNs demonstrate significant improvement in performance. This indicates that the improvement is primarily attributed to the dendritic dynamics rather than simply adding learnable parameters, which can also be supported by Supplementary Figure S7.

Note that the learnable somatic α is termed as the timing factor of membrane potential in our manuscript. In fact, we had clarified in the last sentences of Figure 3&4 captions in the previous manuscript that our vanilla SNNs have already included learnable soma parameters: “*In above experiments, unless otherwise specified, the timing factors of membrane potentials are initialized following a medium distribution and are learnable during training*”.

Comment 2: *All presented results lack standard deviation, which leads me to assume that each network was initialized only once per case. However, given the inherent randomness in both artificial neural networks (ANNs) and spiking neural networks (SNNs), it’s important to demonstrate robustness by reporting not only the best performing model but also the results across multiple initializations. I recommend running the experiments with at least 10 initializations per network and reporting average accuracies with standard deviations (e.g., Figure 4g).*

Response: This is a nice suggestion. In this work, we conducted many experiments in several complex temporal tasks, which are very time-consuming and make the repetitions challenging. To fix this problem raised by the reviewer with acceptable time overheads, we run the experiments with 10 random initializa-

tions in the customized spiking XOR tasks and 5 random initializations on SHD and SCC datasets, and report average accuracies with standard deviations in the revised manuscript. With updated results, our conclusions still hold.

Revision: Added repeated experiments and updated the reported results.

Comment 3: *It is crucial to clarify in the text that the DH-LIF model presented here is not a two-compartment model. Instead, it consists of a soma compartment and a variable number of dendritic branches. This clarification is important as it affects the number of trainable parameters, with each branch having its own α parameter.*

Response: According to this advice, we have replaced “two-compartment” with “multi-compartment” in the revised manuscript.

Revision: Changed “two-compartment” to “multi-compartment”.

Comment 4: *Figure 2 appears to be confusing. It is not immediately evident why the dendritic time constant should match the input frequency. Multi-timescale dynamics arise from various dendritic and synaptic mechanisms, such as ionic channels and NMDARs. I suggest explaining the role of the dendritic time constant in this context and why it is unbounded (within biological ranges). A time scale of approximately 100ms is primarily influenced by NMDA synaptic characteristics rather than internal membrane properties. For instance, in the Methods section, the dendritic time constant is a function of membrane capacitance and resistances, which are structural properties of the dendrite and should be bounded within a biological regime.*

Response: This is a quite professional comment from the biological perspective. We have revised Figure 2 to make it more intuitive and easier to comprehend.

It is true that there are many inherent constraints in biological neurons. However, our work focuses more on appropriately incorporating biological observations into computational models for solving real-world computing tasks, rather than strictly adhering to all biological principles. In fact, many works on bio-inspired algorithms did not follow the strict biological constraints such as the range of timing factors of membrane potential [1], [2] and neuron units communicating through continuous firing rates rather than discrete action potentials [3], otherwise, the model performance usually degrades due to the difficulty in training a model with complicated neural dynamics and biological details. Innovating the learning algorithms might be a promising way to address this problem. In our work, the timing factors are not unbounded and we restrict them within $(0, 1)$ through the $\text{sigmoid}(\cdot)$ function (see Equation (6) in *Methods*), but this is not the result of considering biological constraints. We select this range just because we do not expect negative values when calculating $1 - \alpha$. It is very hard to balance the performance in practical tasks and the biological plausibility, which is an interesting topic for future work.

Revision: Modified Figure 2 to avoid confusion and added discussions on the timing factor constraints in the *Discussion* part of the revised manuscript.

Comment 5: *Writing quality: While I acknowledge the effort put into creating the illustrations and writing the text, the manuscript would benefit from improved language. There are several grammatical and semantic issues that could be easily resolved using grammar-checking software. Additionally, all figure legends should be more descriptive and self-explanatory. Furthermore, the authors employ terminology that can confuse readers (e.g., agile capability, feature fusion, etc.).*

Response: We have polished the manuscript in terms of grammatical and semantic issues, figure legends,

and confusing terminologies. Thanks again for such kind suggestions.

Revision: Polished the manuscript.

REFERENCES

- [1] N. Perez-Nieves, V. C. Leung, P. L. Dragotti, and D. F. Goodman, "Neural heterogeneity promotes robust learning," *Nature communications*, vol. 12, no. 1, pp. 1–9, 2021.
- [2] W. Fang, Z. Yu, Y. Chen, T. Masquelier, T. Huang, and Y. Tian, "Incorporating learnable membrane time constant to enhance learning of spiking neural networks," in *Proceedings of the IEEE/CVF international conference on computer vision*, 2021, pp. 2661–2671.
- [3] D. Sussillo, "Neural circuits as computational dynamical systems," *Current opinion in neurobiology*, vol. 25, pp. 156–163, 2014.

REVIEWERS' COMMENTS

Reviewer #1 (Remarks to the Author):

I highly appreciate the revisions made by the authors. All my comments have been addressed and I believe that this work will contribute to the field of biologically-inspired SNNs for solving complex temporal tasks.

Reviewer #2 (Remarks to the Author):

Thank you for the responses to the previous comments. My comments were fully addressed. I especially appreciate your discussion of the work you had done to look at heterogeneous dendritic connections, even if that work is still inconclusive. I also appreciated the additional comparisons to existing work that were included.

Reviewer #3 (Remarks to the Author):

The authors have replied to all of my comments, and I have no further comments to add.

Minor comments:

1. What is missing is a section with the computational resources used (CPU, GPU, RAM, etc) and also referring to the free libraries they used (e.g, pytorch, python, etc).
2. I have also checked the code; a README is present including useful info, but not explicit steps someone needs to follow in order to reproduce the results. There are only a few inline comments in the code. At a later stage, it would be very useful to add more comments in the code.

Congratulations.

Response Letter

Manuscript ID: NCOMMS-23-35049A

Title: Temporal Dendritic Heterogeneity Incorporated with Spiking Neural Networks for Learning Multi-timescale Dynamics

Authors: Hanle Zheng, Zhong Zheng, Rui Hu, Bo Xiao, Yujie Wu, Fangwen Yu, Xue Liu, Guoqi Li, Lei Deng

We would like to thank the reviewers for spending valuable time and raising relevant comments. The point-by-point responses are provided as follows. All the revisions in the manuscript and Supplementary Information (SI) are marked in blue.

1. Responses to Reviewer 1

Overall Comment: I highly appreciate the revisions made by the authors. All my comments have been addressed and I believe that this work will contribute to the field of biologically-inspired SNNs for solving complex temporal tasks.

Response: We greatly appreciate the reviewer for such generous praise of our work.

2. Response to Reviewer 2

Overall Comment: Thank you for the responses to the previous comments. My comments were fully addressed. I especially appreciate your discussion of the work you had done to look at heterogeneous dendritic connections, even if that work is still inconclusive. I also appreciated the additional comparisons to existing work that were included.

Response: We really thank the reviewer again for the insightful comments provided in the previous review round.

3. Response to Reviewer 3

Overall Comment: The authors have replied to all of my comments, and I have no further comments to add.

Response: We sincerely thank the reviewer very much for the approval for our previous responses.

Minor Comment 1: *What is missing is a section with the computational resources used (CPU, GPU, RAM, etc) and also referring to the free libraries they used (e.g, pytorch, python, etc).*

Response: The codes were built with the Pytorch framework and executed on 8 NVIDIA RTX 3090 GPUs.

Revision: In the revised Methods-Experimental setting Section, added descriptions on the computational resources and the free libraries used.

Minor Comment 2: *I have also checked the code; a README is present including useful info, but not explicit steps someone needs to follow in order to reproduce the results. There are only a few inline comments in the code. At a later stage, it would be very useful to add more comments in the code.*

Response: This is a nice suggestion and we have done.

Revision: Added explicit steps to reproduce the results and more comments in the code. Details can be found in the link <https://github.com/eva1801/DH-SNN>.